# CROSS-DOMAIN RECOMMENDATION FROM IMPLICIT FEEDBACK

## ABSTRACT

Existing *cross-domain recommendation* (CDR) algorithms aim to leverage *explicit feedback* from richer source domains to enhance recommendations in a target domain with limited records. However, practical scenarios often involve easily obtainable *implicit feedback*, such as user clicks, and purchase history, instead of explicit feedback. Thus, in this paper, we consider a more practical problem setting, called *cross-domain recommendation from implicit feedback* (CDRIF), where both source and target domains are based on implicit feedback. We initially observe that current CDR algorithms struggle to make recommendations when implicit feedback *exists* in both source and target domains. The primary issue with current CDR algorithms mainly lies in that implicit feedback can only *approximately* express user preferences in the dataset, inevitably introducing noisy information during the training of recommender systems. To this end, we propose a *noise-aware reweighting framework* (NARF) for CDRIF, which effectively alleviates the negative effects brought by the implicit feedback and improves recommendation performance. Extensive experiments conducted on both synthetic and large real-world datasets demonstrate that NARF, implemented by two representative CDR algorithms, significantly outperforms the baseline methods, which further underscores the significance of handling implicit feedback in CDR. The code is available in an anonymous Github repository: `https://anonymous.4open.science/r/CDR-3E2A/README.md`.

## 1 INTRODUCTION

As the number of online data continues to grow, individuals encounter difficulties in discovering content, products, or services that align with their preferences. To address this challenge, recommendation systems have emerged as a solution, offering users relevant content based on their past behaviour and similarities with other users (Yin et al., 2013; He et al., 2020; Xia et al., 2021). Nonetheless, the majority of these recommender systems struggle to deliver satisfactory content recommendations for new users, particularly those classified as cold-start users, who lack a prior interaction history (Man et al., 2017; Zhu et al., 2022).

Existing *cross-domain recommendation* (CDR) algorithms offer promising solutions to the cold-start problem (Singh & Gordon, 2008). This approach leverages the user's preferences from alternative domains to enhance recommendations in the target domain. Current CDR algorithms are specifically targeted to handle *explicit feedback* (Man et al., 2017; Zhu et al., 2021; 2022), which directly reflects users' preferences. In platforms such as MovieLens, Netflix, and Amazon Review, users explicitly indicate their preferences. Explicit feedback contains user-item interaction with positive preference (i.e. positive pairs) and negative preference (i.e. negative pairs).

However, as noted by Hu et al. (2008), explicit feedback may not consistently be accessible in real-world scenarios, and thereby recommender systems can extract user preferences from a more abundant source of easily collectible *implicit feedback*, which is not explicitly provided by users but is derived from their interactions or usage patterns. For instance, an e-commerce website can collect customers' purchase histories to infer their preferences and provide tailored recommendations. Similarly, movie websites can suggest new films to users by considering the number of views, rather than heavily relying on explicit preference information.

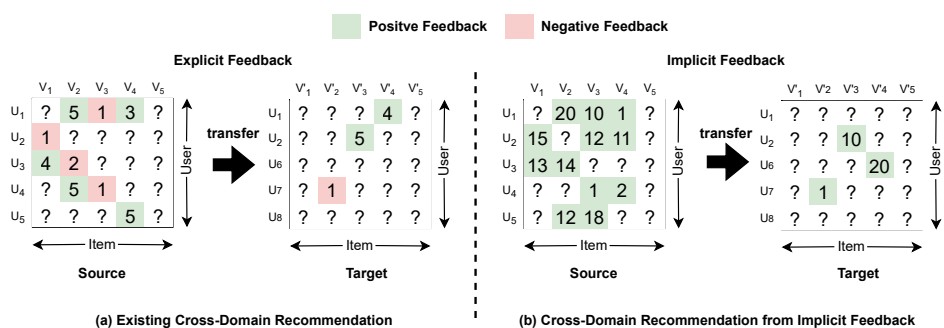

Figure 1: The *cross-domain recommendation from implicit feedback* (CDRIF) and existing *cross-domain recommendation* (CDR) problem settings. In CDRIF, the objective is to perform CDR using only implicit feedback data. This scenario aligns closely with real-world situations where users may be reluctant to provide explicit ratings. Therefore, CDRIF is a more challenging and practical problem setting compared to traditional CDR which relies on explicit feedback.

Table 1: Comparison results of direct use of CDR algorithm (PTUPCDR) and ours on the real-world implicit feedback dataset. *Difference* denotes the relative performance difference. Higher values of these metrics represent better performance in recommendation.

| Methods | Recall@50 | Recall@100 | NDCG@10 | NDCG@100 |
|---|---|---|---|---|
| PTUPCDR (Zhu et al., 2022) | 0.0052 | 0.0101 | 0.0033 | 0.0055 |
| Ours | 0.1268 | 0.1767 | 0.1066 | 0.1275 |
| Difference | 2338.5% | 1649.5% | 3130.3% | 2218.2% |

Thus, in this paper, we consider a more practical problem setting, called *cross-domain recommendation from implicit feedback* (CDRIF), where both source and target domains consist of implicit feedback (see Figure 1). To the best of our knowledge, prior to our study, no existing CDR research has thoroughly explored *the unique challenges inherent in implicit feedback*. The advantage of CDRIF lies in its ability to address scenarios in which users may be hesitant to provide explicit ratings or where system limitations prevent the collection of explicit feedback (Hu et al., 2008). Furthermore, implicit feedback is easier to collect and tends to be less sparse than explicit feedback, as it can be passively generated by users. This characteristic aligns closely with real-world scenarios, making it a valuable resource for recommendation systems (Hu et al., 2008; Lu et al., 2018; Wen et al., 2019).

To address CDRIF, one straightforward solution is to employ existing CDR algorithms. However, we have observed that current CDR algorithms fail to recommend when implicit feedback *exists* in source and target domains. As illustrated in Table 1, the *Difference* row highlights the substantial relative improvement of the *Ours* method over the *Existing CDR* method, with percentage differences ranging from approximately 1650% to 3130% across the four metrics, indicating significantly better performance by *Ours*. The failure of current CDR algorithms mainly lies in that implicit feedback can only provide an *approximate* representation of users' preference within the dataset, which inevitably introduces the noise information during the training of the recommender systems.

There are two challenges when considering implicit feedback in CDR. The first challenge arises from the absence of negative signals in implicit feedback (Hu et al., 2008). To address this issue, there are two main methods: sampling and non-sampling methods, both of which will introduce noisy data. For example, sampled negative data may unintentionally include items that users are interested in. Similarly, non-sampling methods hold the assumption that all unobserved data represents items the user is not interested in, which may not accurately reflect real-world scenarios.

The second challenge is that the numerical value of implicit feedback signifies *confidence* rather than *preference* (Hu et al., 2008). Those numerical values of implicit feedback represent the frequency of actions, e.g., the number of times a user has purchased an item in an e-commerce platform, etc., while they do not inherently indicate the user's preference. To illustrate, a considerable number of clicks might not necessarily lead to purchases, and numerous purchases might result in negative reviews (Wang et al., 2021b). Implicit feedback is inherently noisy, and prior studies (Hu et al., 2008; Lu et al., 2018; Wen et al., 2019) have highlighted the disparity between implicit feedback and the actual user satisfaction, attributing this to the prevalence of noisy interactions. Hence, both challenges posed by implicit feedback can *underperform* the existing CDR algorithms if left unad-

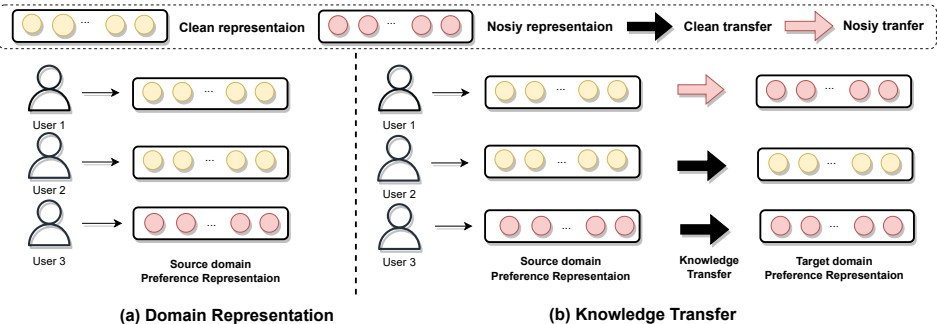

Figure 2: Challenges brought by implicit feedback in *cross-domain recommendation* (CDR) are depicted as follows: (a) Noisy data within implicit feedback can impact the representation of domain preferences; (b) Knowledge transfer can result in a noisy target representation when the source representation is trained on noisy data. Moreover, even a clean representation may still introduce noise during transfer, as the learning of knowledge transfer is influenced by noisy labelled data.

dressed in the context of CDRIF. An inherently noisy dataset has the potential to hinder the effective learning of domain representation and knowledge transfer, as described in Figure 2.

To address the challenges, we propose a *noise-aware reweighting framework* (NARF) for CDRIF (as shown in Figure 3), which significantly alleviates the negative effects arising from implicit feedback. NARF contains two essential steps, effectively addressing the challenges mentioned above: (i) *Implicit feedback calibration* (IFC): This initial step aims to calibrate the training data to better reflect users' actual preferences. Specifically, it involves determining the calibration factor for each user-item pair, with a higher value indicating closer alignment with the user's genuine preferences. Based on these factors, we can prioritize training the recommender system with data that carry higher factors. (ii) *Dynamic noise reduction* (DNR): The second step, DNR, is designed to dynamically reduce the noise present in the data during the learning process. Noise in the data can lead to poor model generalization and recommendation quality. DNR likely involves strategies for identifying and eliminating noisy data points, thereby improving the model's robustness to noise.

Extensive experiments conducted on both synthetic and large real-world datasets show that NARF as implemented by two representative CDR algorithms, consistently and significantly outperforms the baseline methods, which further confirms the significance of handling implicit feedback in CDR. Furthermore, this paper dives into a discussion regarding the mechanisms behind the improvements achieved by NARF. It aims to provide insights into how NARF effectively tackles the challenges posed by implicit feedback in CDRIF scenarios. By shedding light on these explanations, this research not only highlights the significance of NARF but also contributes valuable knowledge to the broader field of recommender systems.

## 2 PROBLEM SETTING

**Basic Notations.** We first introduce some fundamental notations in conventional recommendation problems. Let $\mathbf{R} \in \mathbb{R}^{m \times n}$ represent a feedback matrix in which its elements are expressed by $r_{ij}$ ($i = 1, \ldots, m$ and $j = 1, \ldots, n$). $\mathbb{U} = \{u_1, u_2, \ldots, u_m\}$ and $\mathbb{V} = \{v_1, v_2, \ldots, v_n\}$ denotes as the user and item sets in the feedback matrix $\mathbf{R}$, where $m$ and $n$ are the number of users and items, respectively. $r_{ij}$ represents an interaction between user $u_i$ and item $v_j$, which could take the form of explicit feedback (e.g., ratings) or implicit feedback (e.g., number of clicks). We also introduce a set $\mathbb{D}_{\text{obv}} = \{(u_i, v_j, r_{ij}) \mid u_i \in \mathbb{U}, v_j \in \mathbb{V}, r_{ij} \neq 0\}$ to represent the set of observed entries (i.e., $r_{ij} \neq 0$) in $\mathbf{R}$ and a set $\mathbb{D}_{\text{unk}} = \{(u_i, v_j) \mid u_i \in \mathbb{U}, v_j \in \mathbb{V}, r_{ij} = 0\}$ to represent the set of unknown entries (i.e., $r_{ij} = 0$) in $\mathbf{R}$. In conventional recommendation problems, we aim to train a *recommendation function* $f : \mathbb{U} \times \mathbb{V} \to \mathbb{R}$ with $\mathbb{D}_{\text{obv}}$ and use $f$ to predict the interaction between $u_i$ and $v_j$ in $\mathbb{D}_{\text{unk}}$.

**CDRIF Problem.** Then, we can outline the CDRIF in the following. In CDRIF, we have $\mathbf{R}^{\text{s}}$ and $\mathbf{R}^{\text{t}}$ to represent two implicit feedback matrices corresponding to the source and target domains, respectively. Namely, $r_{ij}^{\text{s}}$ in $\mathbf{R}^{\text{s}}$ and $r_{ij}^{\text{t}}$ in $\mathbf{R}^{\text{t}}$ can only represent the weak signals of the preference between users and items. We also have $\mathbb{U}^{\text{s}} = \{u_1^{\text{s}}, u_2^{\text{s}}, \ldots, u_m^{\text{s}}\}$ and $\mathbb{V}^{\text{s}} = \{v_1^{\text{s}}, v_2^{\text{s}}, \ldots, v_n^{\text{s}}\}$ to repre-

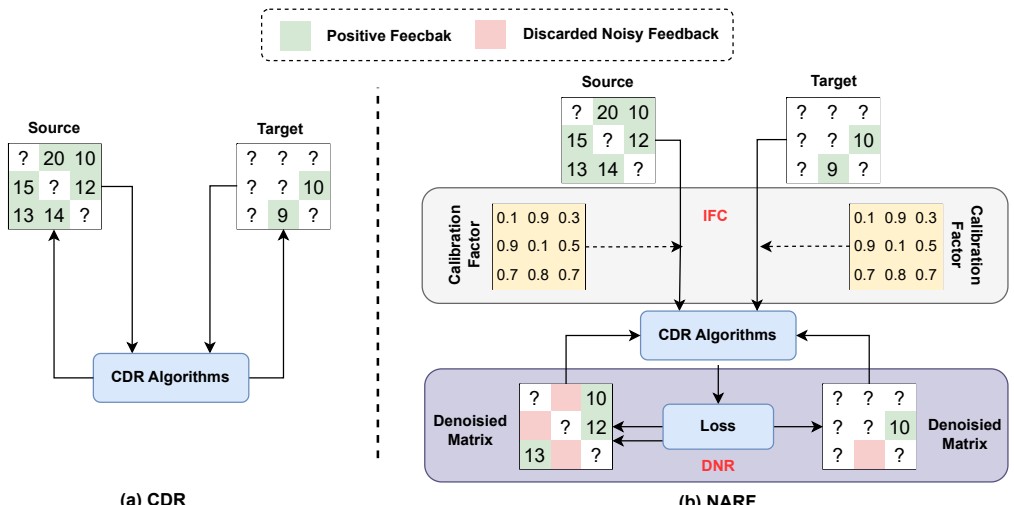

Figure 3: The comparison among (a) *cross-domain recommendation* (CDR), and (b) *noise-aware reweighting framework* (NARF), which consists of two main strategies: implicit feedback calibration (IFC) and dynamic noise reduction (DNR). It is worth noting that, for simplicity, we have omitted the negative sampling step.

sent user and item sets in the source domain and $\mathbb{U}^{\mathrm{t}} = \{u_1^{\mathrm{t}}, u_2^{\mathrm{t}}, \ldots, u_m^{\mathrm{t}}\}$ and $\mathbb{V}^{\mathrm{t}} = \{v_1^{\mathrm{t}}, v_2^{\mathrm{t}}, \ldots, v_n^{\mathrm{t}}\}$ to represent user set and item sets in the target domain, where $\mathbb{U}^{\mathrm{o}} = \mathbb{U}^{\mathrm{s}} \cap \mathbb{U}^{\mathrm{t}} \neq \emptyset$ and $\mathbb{V}^{\mathrm{s}} \cap \mathbb{V}^{\mathrm{t}} = \emptyset$ (i.e., two domains only have overlapping users).

For both domains, we have observed entries as well as unknown entries, i.e., we have $\mathbb{D}_{\mathrm{obv}}^{\mathrm{s}}$ and $\mathbb{D}_{\mathrm{unk}}^{\mathrm{s}}$ for the source domain and $\mathbb{D}_{\mathrm{obv}}^{\mathrm{t}}$ and $\mathbb{D}_{\mathrm{unk}}^{\mathrm{t}}$ for the target domain. In CDRIF, we want to train a *recommendation function* $f : \mathbb{U}^{\mathrm{t}} \times \mathbb{V}^{\mathrm{t}} \to \mathbb{R}^+$ with $\mathbb{D}_{\mathrm{obv}}^{\mathrm{s}}$ and $\mathbb{D}_{\mathrm{obv}}^{\mathrm{t}}$ and use the trained $f$ to predict interaction between $u_i^{\mathrm{t}}$ and $v_j^{\mathrm{t}}$ in $\mathbb{D}_{\mathrm{unk}}^{\mathrm{t}}$.

**Remark 1** *As this is the first exploration of CDRIF, we only consider a scenario where only user-item interactions are accessible when training recommender systems. Incorporating additional information regarding users or items, such as dwell time and gaze pattern, might enhance the recommendation accuracy but falls outside the scope of our paper. Our primary objective is to study how implicit feedback influences CDR and devise strategies to mitigate its inherent noise.*

## 3 PRELIMINARY

**Learning from Implicit Data.** The recommendation problem with implicit feedback is formulated as the problem of estimating the scores of unobserved entries in the feedback matrix $\mathbf{R}$. Following Hu et al. (2008) and He et al. (2017), a set of binary variables is defined as $y_{ij} = \mathbb{1}_{r_{ij}>0}(r_{ij})$, where $\mathbb{1}_{r_{ij}>0}(r_{ij})$ is an indicator function. It yields a value of 1 if $r_{ij} > 0$, indicating the presence of an interaction between user $u_i$ and item $v_j$, and 0 otherwise. As for model training, to learn a recommendation function $f$ (defined in Section 2) with parameters $\Theta$, the optimizer minimizes the training loss as $\min \mathcal{L}(\Theta, \widetilde{\mathbb{D}}_{\mathrm{obv}})$, where $\widetilde{\mathbb{D}}_{\mathrm{obv}} = \{(u_i, v_j, y_{ij}) \mid u_i \in \mathbb{U}, v_j \in \mathbb{V}\}$ denotes the training data fed into the model, $y_{ij}$ represent the pre-processed label according to $y_{ij}$, and $\mathcal{L}$ is the average loss over $\widetilde{\mathbb{D}}_{\mathrm{obv}}$. The log loss (He et al., 2017; Guo et al., 2017; Song et al., 2019) is commonly used as a loss function in recommendation tasks. More related works are demonstrated in Appendix A.1.

**Matrix Factorization.** Given the feedback matrix $\mathbf{R} \in \mathbb{R}^{m \times n}$, two embedding models $\phi : \mathbb{U} \to \mathbb{R}^d$ and $\psi : \mathbb{V} \to \mathbb{R}^d$ are employed to learn user and item embeddings with dimension $d$. Matrix factorization (MF) (Koren, 2008) estimates the score between the $i$-th user and $j$-th item using the dot product of their embeddings as $\hat{r}_{ij} = \langle \phi(u_i), \psi(v_j) \rangle$.

**CDR Algorithm.** A group of recent CDR methods, which emphasize the alignment of user preferences (Man et al., 2017; Bi et al., 2020; Wang et al., 2021a; Zhu et al., 2021; 2022) typically involves two main steps: (i) latent factor modelling, and (ii) latent space mapping. This initial step focuses on learning embedding functions for both the source and target domains, denoted as $\phi^{\mathrm{s}}, \psi^{\mathrm{s}}, \phi^{\mathrm{t}}, \psi^{\mathrm{t}}$.

During the latent space mapping, the aim is to train a mapping function $\rho : \mathbb{R}^d \times \cup_{i=1}^n \mathbb{R}^{i \times d} \to \mathbb{R}^d$ parameterised by $\Theta$. The $\rho(\cdot)$ is designed to establish the relationships between the latent space of domains. Importantly, the specific form of $\rho(\cdot)$ can vary among different CDR methods. Once the function $\rho(\cdot)$ is selected, the optimized parameters of $\rho(\cdot)$ is

$$\Theta^* = \arg\min_{\Theta} \frac{1}{|\mathbb{S}^o|} \sum_{(u_i, v_j, r_{ij}, \mathbb{V}_i^s) \in \mathbb{S}^o} \ell(r_{ij}, \langle \rho(\phi^s(u_i), \Psi_{\mathbb{V}_{u_i}^s}^s; \Theta), \psi^t(v_j) \rangle), \tag{1}$$

where $\mathbb{S}^o = \{(u_i, v_j, r_{ij}, \mathbb{V}_{u_i}^s) | u_i \in \mathbb{U}^o, (u_i, v_j, r_{ij}) \in \mathbb{D}_{\text{obv}}^t\}$, $\Psi_{\mathbb{V}_{u_i}^s}^s = \{\psi^s(v) | v \in \mathbb{V}_{u_i}^s\}$, and $\ell$ denotes the loss function. $\mathbb{V}_{u_i}^s$ is defined as

$$\{v_j \in \mathbb{V}^s | r_{ij} > 0, (u_i, v_j, r_{ij}) \in \mathbb{D}_{\text{obv}}^s\}, \tag{2}$$

which represents a list of source-domain items $v_j$ interacted with the $i$-th user who is overlapping in both domains (i.e., $r_{ij}^s > 0$).

## 4 NOISE-AWARE REWEIGHTING FRAMEWORK

This section introduces *noise-aware reweighting framework* (NARF), a general framework designed to enhance the effectiveness of learning in CDRIF.

**Implicit Feedback Calibration (IFC).** Given two sets of observed entries, denoted as $\mathbb{D}_{\text{obv}}^s$ for the source domain and $\mathbb{D}_{\text{obv}}^t$ for the target domain, where each pair is represented as $(u_i, v_j, r_{ij})$ with $u_i$ belonging to either $\mathbb{U}^s$ or $\mathbb{U}^t$, $v_j$ belonging to $\mathbb{V}^s$ or $\mathbb{V}^t$, and $r_{ij}$ is a non-zero value. In IFC, we first employ a predefined sampling strategy denoted as $S_k : \mathbb{D} \to \mathcal{P}_k(\mathbb{D})$ to sample a subset $\mathbb{D}_{\text{unk}}^{s,k}$ from $\mathbb{D}_{\text{unk}}^s$ and a subset $\mathbb{D}_{\text{unk}}^{t,k}$ from $\mathbb{D}_{\text{unk}}^t$. $\mathcal{P}_k(\mathbb{D}) = \{\mathbb{D}_k \subset \mathbb{D} | |\mathbb{D}_k| = k\}$ consists of subsets of $\mathbb{D}$ whose cardinality is $k$. Namely, the cardinalities of $\mathbb{D}_{\text{unk}}^{s,k}$ and $\mathbb{D}_{\text{unk}}^{t,k}$ are $k$. Formally, based on $S_k$ defined above, we know

$$\mathbb{D}_{\text{unk}}^{s,k} = S_k(\mathbb{D}_{\text{unk}}^s), \ \mathbb{D}_{\text{unk}}^{t,k} = S_k(\mathbb{D}_{\text{unk}}^t). \tag{3}$$

Given that the preferences of user-item pairs in $\mathbb{D}_{\text{unk}}^{s,k}$ and $\mathbb{D}_{\text{unk}}^{t,k}$ are not observed in historical data, we treat them as negative pairs in IFC. Specifically, we can have two new sets to represent the negative preference relationships between users and items:

$$\widetilde{\mathbb{D}}_{\text{unk}}^{s,k} = \{(u_i, v_j, 0) | (u_i, v_j) \in S_k(\mathbb{D}_{\text{unk}}^s)\}, \ \widetilde{\mathbb{D}}_{\text{unk}}^{t,k} = \{(u_i, v_j, 0) | (u_i, v_j) \in S_k(\mathbb{D}_{\text{unk}}^t)\}, \tag{4}$$

where we assign a "0" feedback value to all user-item pairs in $\mathbb{D}_{\text{unk}}^{s,k}$ and $\mathbb{D}_{\text{unk}}^{t,k}$. By using $\widetilde{\mathbb{D}}_{\text{unk}}^{s,k}$ and $\widetilde{\mathbb{D}}_{\text{unk}}^{t,k}$, we can generate two new training sets, denoted as $\widetilde{\mathbb{D}}^s$ and $\widetilde{\mathbb{D}}^t$, which now include sampled negative entries:

$$\widetilde{\mathbb{D}}^s = \mathbb{D}_{\text{obv}}^s \cup \widetilde{\mathbb{D}}_{\text{unk}}^{s,k}, \ \widetilde{\mathbb{D}}^t = \mathbb{D}_{\text{obv}}^t \cup \widetilde{\mathbb{D}}_{\text{unk}}^{t,k}. \tag{5}$$

Building upon the analysis of implicit feedback presented in Section 1, it is evident that $\mathbb{D}_{\text{obv}}^s$, $\mathbb{D}_{\text{obv}}^t$, $\widetilde{\mathbb{D}}_{\text{unk}}^{s,k}$ and $\widetilde{\mathbb{D}}_{\text{unk}}^{t,k}$ contain noisy information and can only approximately express the users' true preferences. Thus, it becomes essential to introduce *calibration factors* to assess whether an element in $\widetilde{\mathbb{D}}^s$ or $\widetilde{\mathbb{D}}^t$ can accurately reflect a users' true preference. Formally, a calibration factor is a function $c : \mathbb{U} \times \mathbb{V} \times \mathbb{R} \to \mathbb{R}^+$, which maps an element $s$ in $\widetilde{\mathbb{D}}^s$ or $\widetilde{\mathbb{D}}^t$ to a positive real value. A higher calibration factor indicates that $s$ is more capable of reflecting the user's preference accurately. With the calibration factor function $c$ in place, we can obtain two sets of calibration factors for $\widetilde{\mathbb{D}}^s$ and $\widetilde{\mathbb{D}}^s$:

$$\widetilde{\mathbb{K}}^s = \{c(u_i, v_j, r_{ij}) | (u_i, v_j, r_{ij}) \in \widetilde{\mathbb{D}}^s\}, \ \widetilde{\mathbb{K}}^t = \{c(u_i, v_j, r_{ij}) | (u_i, v_j, r_{ij}) \in \widetilde{\mathbb{D}}^t\}. \tag{6}$$

$\widetilde{\mathbb{K}}^s$ and $\widetilde{\mathbb{K}}^t$ can provide a guide to find important elements $s$ in $\widetilde{\mathbb{D}}^s$ and $\widetilde{\mathbb{D}}^t$ (see the next part).

**Remark 2** *Note that, based on $\widetilde{\mathbb{K}}^s$ and $\widetilde{\mathbb{K}}^t$, we have the ability to integrate them into common CDR methods via assigning high weights to elements with high calibration factors during training (see Eq.* (1)*). In experiments, we name this type of method as "X-NARF-I" where "X" represents a CDR method (e.g., "X" could be the EMCDR method). Based on the experimental results, "X-NARF-I" has demonstrated great improvements compared to baseline methods.*

We inject calibration factors $\widetilde{\mathbb{K}}^{\mathrm{s}}$, $\widetilde{\mathbb{K}}^{\mathrm{t}}$ into $\widetilde{\mathbb{D}}^{\mathrm{s}}$, $\widetilde{\mathbb{D}}^{\mathrm{t}}$, obtaining training sets $\mathbb{S}^{\mathrm{s}}$ and $\mathbb{S}^{\mathrm{t}}$ for the source and target domain embedding learning, and $\mathbb{S}^{\mathrm{o}}$ for mapping function learning as

$$\mathbb{S}^{\mathrm{s}} = \{(u_i, v_j, r_{ij}, k_{ij}) | (u_i, v_j, r_{ij}) \in \widetilde{\mathbb{D}}^{\mathrm{s}}, k_{ij} \in \widetilde{\mathbb{K}}^{\mathrm{s}}\}, \tag{7}$$

$$\mathbb{S}^{\mathrm{t}} = \{(u_i, v_j, r_{ij}, k_{ij}) | (u_i, v_j, r_{ij}) \in \widetilde{\mathbb{D}}^{\mathrm{t}}, k_{ij} \in \widetilde{\mathbb{K}}^{\mathrm{t}}\}, \tag{8}$$

$$\mathbb{S}^{\mathrm{o}} = \{(u_i, v_j, r_{ij}, \mathbb{V}^{\mathrm{s}}_{u_i}), k_{ij} | u_i \in \mathbb{U}^{\mathrm{o}}, (u_i, v_j, r_{ij}) \in \widetilde{\mathbb{D}}^{\mathrm{t}}, k_{ij} \in \widetilde{\mathbb{K}}^{\mathrm{t}}\}, \tag{9}$$

where $\mathbb{V}^{\mathrm{s}}_{u_i}$ is defined in Eq. (2).

**Dynamic Noise Reduction (DNR).** Although we employ a calibration factor function to assess whether an entry in $\widetilde{\mathbb{D}}^{\mathrm{s}}$ or $\widetilde{\mathbb{D}}^{\mathrm{t}}$ can accurately reflect the user's true preference, it is important to acknowledge that this function may still include noisy information. Thus, we introduce DNR to dynamically eliminate unreliable user-item pairs during the training process. DNR contains two stages: (i) **Learning embeddings** of users and items in both domains denoted as $\phi^{\mathrm{s}*}, \psi^{\mathrm{s}*}, \phi^{\mathrm{t}*}, \psi^{\mathrm{t}*}$; and (ii) **Learning mapping** function $\Theta^*$. During the first stage of DNR, based on the loss values within each mini-batch $\mathbb{B}$ during the $T$-th training epoch, we can obtain reliable mini-batch data as

$$\tilde{\mathbb{B}} = \underset{\mathbb{B}':|\mathbb{B}'|\geq R(T)|\mathbb{B}|}{\arg\min} \mathcal{L}(\ell, \mathbb{B}'), \tag{10}$$

where $R(T)$ determines the number of instances with small loss that are considered reliable for training at round $T$, and $\mathcal{L}$ calculates the average loss for mini-batch $\mathbb{B}'$ using the loss function $\ell$, where $\ell$ is weighted by $k_{ij}$. Then we use the filtered set $\tilde{\mathbb{B}}$ to train a more reliable user and item embedding as follows

$$\tilde{\phi}, \tilde{\psi} = \underset{\phi,\psi}{\arg\min} \frac{1}{|\tilde{\mathbb{B}}|} \sum_{(u_i, v_j, r_{ij}, k_{ij}) \in \tilde{\mathbb{B}}} k_{ij} \cdot \ell(r_{ij}, \langle \phi(u_i), \psi(v_j) \rangle), \tag{11}$$

In stage two of DNR, we similarly use Eq. (10) to obtain reliable mini-batch data and then the optimized parameters $\Theta$ are

$$\tilde{\Theta} = \underset{\Theta}{\arg\min} \frac{1}{|\tilde{\mathbb{B}}^{\mathrm{o}}|} \sum_{(u_i, v_j, r_{ij}, \mathbb{V}^{\mathrm{s}}_i, k_{ij}) \in \tilde{\mathbb{B}}^{\mathrm{o}}} k_{ij} \cdot \ell(r_{ij}, \langle \rho(\phi^{\mathrm{s}*}(u_i), \Psi^{\mathrm{s}}_{\mathbb{V}^{\mathrm{s}}_{u_i}}; \Theta), \psi^{\mathrm{t}*}(v_j) \rangle). \tag{12}$$

Algorithm 1 summarizes the entire NARF procedure and provides the output of $\rho(; \Theta^*), \phi^{\mathrm{s}*}, \psi^{\mathrm{s}*}, \phi^{\mathrm{t}*}$ and $\psi^{\mathrm{t}*}$. It is important to note that NARF serves as a flexible framework that can be implemented with various methods. For example, the function $c$ can be tailored to different scenarios to suit specific needs; Eq. (10) can be realized by existing denoising algorithms; The mapping function $\rho$ can be employed from existing CDR methods. At the end of this section, we will introduce the proper calibration function $c$, Eq. (10), and the mapping function $\rho$.

**Inference Procedure of NARF.** To predict the preferences of a cold-start target-domain user $u$ who has interactions in the source domain and a target-domain item $v$, we can utilize the outputs from NARF. Specifically, we can obtain the embedding of user $u$ as follows:

$$\rho(\phi^{\mathrm{s}*}(u), \Psi^{\mathrm{s}*}_{\mathbb{V}^{\mathrm{s}}_u}; \Theta^*), \text{ where } \Psi^{\mathrm{s}}_{\mathbb{V}^{\mathrm{s}}_u} = \{\psi^{\mathrm{s}*}(v) | v \in \mathbb{V}^{\mathrm{s}}_u\}, \tag{13}$$

and $\mathbb{V}^{\mathrm{s}}_u$ is obtained by Eq. (2). Thus, the preference of the user $u$ on an item $v$ is predicted by $\langle \rho(\phi^{\mathrm{s}*}(u), \Psi^{\mathrm{s}}_{\mathbb{V}^{\mathrm{s}}_u}; \Theta^*), \psi^{\mathrm{t}*}(v_j) \rangle$.

**Understanding of NARF.** Since implicit feedback lacks negative signals for users, it poses a challenge to determine negative preferences from the implicit feedback. In IFC, our goal is to sample negative pairs from the available unobserved data to represent the users' negative preferences. The calibration factor function is employed to better align with users' actual preferences, guiding the identification of reliable user-item pairs, which is crucial for efficient CDR learning. It is worth noting that the specific calibration method employed can vary depending on the dataset, as our prior knowledge of the data. However, it is observed that despite these efforts, noisy data persists and has a detrimental impact on model performance.

We recognize that the process of negative sampling in IFC is susceptible to producing data with noisy labels. For instance, selecting data as negative pairs that the user may like could inadvertently impact the model's performance. Similarly, positive-labeled pairs may also contain instances of

---

**Algorithm 1** Noise-aware Reweighting Framework for CDRIF

---

**Input** : observed and unknown sets $\mathbb{D}_{\text{obv}}^{\text{s}}, \mathbb{D}_{\text{obv}}^{\text{t}}, \mathbb{D}_{\text{unk}}^{\text{s}}, \mathbb{D}_{\text{unk}}^{\text{t}}$; sampling strategy $S_k$; calibration factor function
$c$; denoising function $R$; loss function $\ell$; embedding models $\phi^{\text{s}}, \psi^{\text{s}}, \phi^{\text{t}}, \psi^{\text{t}}$; mapping function $\rho(; \Theta)$

`# IFC: Computing calibration factors`
**1: Sample** a subset with sampling strategy $S_k$ as $\mathbb{D}_{\text{unk}}^{\text{s},k}, \mathbb{D}_{\text{unk}}^{\text{t},k}$ from $\mathbb{D}_{\text{unk}}^{\text{s}}, \mathbb{D}_{\text{unk}}^{\text{t}}$ according to Eq. (4);

**2: Combine** $\mathbb{D}_{\text{unk}}^{\text{s},k}, \mathbb{D}_{\text{unk}}^{\text{t},k}$ with $\mathbb{D}_{\text{obv}}^{\text{s}}, \mathbb{D}_{\text{obv}}^{\text{t}}$ to obtain $\widetilde{\mathbb{D}}^{\text{s}}, \widetilde{\mathbb{D}}^{\text{t}}$ according to Eq. (5);

**3: Compute** calibration factors $\widetilde{\mathbb{K}}^{\text{s}}, \widetilde{\mathbb{K}}^{\text{t}}$ according to the function $c$;      *# c can be implemented by Eq. (14)*

**4: Inject** calibration factors $\widetilde{\mathbb{K}}^{\text{s}}, \widetilde{\mathbb{K}}^{\text{t}}$ into $\widetilde{\mathbb{D}}^{\text{s}}, \widetilde{\mathbb{D}}^{\text{t}}$ to obtain training sets as $\mathbb{S}^{\text{s}}, \mathbb{S}^{\text{t}}, \mathbb{S}^{\text{o}}$ according to Eq. (7);

`# DNR Stage One - Learning user and item embeddings` $\phi^{\text{s}*}, \psi^{\text{s}*}, \phi^{\text{t}*}, \psi^{\text{t}*}$
**for** $\mathbb{S}$ in $\{\mathbb{S}^{\text{s}}, \mathbb{S}^{\text{t}}\}$ **do**
    **for** $T = 1$ **to** $T_{\max}$ **do**
        **5: Fetch** mini-batch $\mathbb{B}$ from $\mathbb{S}$
        **6: Obtain** reliable $\widetilde{\mathbb{B}}$ according to Eq. (10);        *# Eq. (10) can be implemented by AD or CTD*
        **7: Update** $\tilde{\phi}, \tilde{\psi}$ according to Eq. (11);
    **end**
**end**

`# DNR Stage Two - Learning a mapping function` $\rho(\cdot; \Theta^*)$
**for** $T = 1$ **to** $T_{\max}$ **do**
    **8: Fetch** mini-batch $\mathbb{B}^{\text{o}}$ from $\mathbb{S}^{\text{o}}$
    **9: Obtain** reliable $\widetilde{\mathbb{B}}^{\text{o}}$ according to Eq. (10);        *# Eq. (10) can be implemented by AD or CTD*
    **10: Update** $\tilde{\Theta}$ according to Eq. (12) and $\rho$;     *# ρ can be a mapping function in existing CDR methods*
**end**

**Output:** the optimized parameters $\rho(\cdot; \Theta^*), \phi^{\text{s}*}, \psi^{\text{s}*}, \phi^{\text{t}*}, \psi^{\text{t}*}$

---

noisy data. This occurs because the feedback value does not always indicate true user preference. Consequently, we need to handle this noisy information by DNR. Experimental results show that IFC and DNR can greatly improve the recommendation performance.

**Realization of NARF: IFC.** To fetch the negative pairs from the feedback matrix, $S_k$ is defined as $S_k = \text{Uniform}(\cdot)$ (Rendle et al., 2012; Ding et al., 2020), which is a widely used sampling strategy (Chen et al., 2023). Additionally, we design a calibration factor function $c$ as

$$c(u_i, v_j, r_{ij}; \boldsymbol{p}) = \underbrace{\mathbb{1}_{r_{ij}>0}(r_{ij}) \cdot (\alpha \cdot r_{ij})}_{\text{part (i): positive pair}} + \underbrace{(1 - \mathbb{1}_{r_{ij}>0}(r_{ij})) \cdot \left(1 + \beta \cdot \frac{p_j}{\sum_{k=1}^{n} p_k}\right)}_{\text{part (ii): negative pair}}, \qquad (14)$$

where $\boldsymbol{p} = [p_1, \ldots, p_n]$, each element $p_j$ in $\boldsymbol{p}$ represents total interactions of the $j$-th item in a dataset. From the definition of $p_j$, we know that the $j$-th item is more popular if the value of $p_j$ is higher. Based on this observation, we can further compute the calibration factors for negative pairs: if $p_j$ is higher, then this negative pair is closer to reflecting the user's true preference, which is expressed in the part (ii) of Eq. (14). As for part (i) of Eq. (14), $\alpha$ is a hyperparameter to control the rate of increase of confidence for positive pairs.

**Realization of NARF: DNR.** To dynamically denoise implicit feedback and learn a good embedding function for new users, we employ two methods, *adaptive denoising* (AD) (Wang et al., 2021b) and *co-teaching denoising* (CTD) (Han et al., 2018), to implement Eq. (10). Then we adopt $\rho$ used in EMCDR (Man et al., 2017) and PTUPCDR (Zhu et al., 2022) in our NARF.

## 5   EXPERIMENTS

**Dataset and Experimental Setups.** We conducted experiments on two synthetic tasks based on Amazon reviews and *one real-world task* collected by ourselves. For the Amazon review dataset, we focus on three popular categories: movies_and_tv, cds_and_vinyl, and books. Two CDR tasks are defined: Task 1 (Movie to Music) and Task 2 (Book to Movie). The dataset was adapted by filtering ratings, adjusting scores, and introducing label noise to simulate CDRIF scenarios. More importantly, two larger real-world datasets were collected from PubMed and DBLP, for Task 3 (PubMed to DBLP), to evaluate our proposed method in real-world scenarios. The statistics of experimental tasks in CDRIF are shown in Table 5. Refer to Appendix A.2 for more details.

Table 2: Evaluation of methods in CDRIF for Task 1. The highest scores are highlighted in bold. $\epsilon$ indicates the noise level in the data. Furthermore, we use *improve.* to indicate the improvement achieved by our best method over the best baseline.

| Method | $\epsilon = 10\%$ | | | | $\epsilon = 15\%$ | | | | $\epsilon = 20\%$ | | | |
|---|---|---|---|---|---|---|---|---|---|---|---|---|
| | R@50 | R@100 | N@50 | N@100 | R@50 | R@100 | N@50 | N@100 | R@50 | R@100 | N@50 | N@100 |
| EMCDR | 0.0102 | 0.0200 | 0.0106 | 0.0143 | 0.0085 | 0.0172 | 0.0089 | 0.0124 | 0.0075 | 0.0133 | 0.0086 | 0.0105 |
| PTUPCDR | 0.0124 | 0.0213 | 0.0143 | 0.0174 | 0.0119 | 0.0212 | 0.0112 | 0.0148 | 0.0086 | 0.0166 | 0.0089 | 0.0119 |
| E-LID | 0.0192 | 0.0307 | 0.0202 | 0.0248 | 0.0159 | 0.0270 | 0.0178 | 0.0217 | 0.0128 | 0.0224 | 0.0136 | 0.0173 |
| P-LID | 0.0201 | 0.0338 | 0.0220 | 0.0270 | 0.0154 | 0.0276 | 0.0170 | 0.0213 | 0.0120 | 0.0222 | 0.0139 | 0.0177 |
| E-NARF-I | 0.0336 | 0.0562 | 0.0329 | 0.0419 | 0.0288 | 0.0470 | 0.0300 | 0.0367 | 0.0239 | 0.0423 | 0.0272 | 0.0338 |
| E-NARF-IA | 0.0429 | 0.0717 | 0.0483 | 0.0587 | 0.0384 | 0.0649 | 0.0429 | 0.0520 | 0.0337 | 0.0557 | 0.0399 | 0.0471 |
| E-NARF-IC | 0.0448 | 0.0726 | **0.0505** | **0.0600** | **0.0418** | 0.0680 | **0.0461** | **0.0552** | 0.0352 | 0.0570 | 0.0400 | 0.0472 |
| P-NARF-I | 0.0331 | 0.0531 | 0.0338 | 0.0410 | 0.0286 | 0.0491 | 0.0294 | 0.0374 | 0.0249 | 0.0415 | 0.0260 | 0.0324 |
| P-NARF-IA | 0.0408 | 0.0701 | 0.0457 | 0.0562 | 0.0383 | 0.0626 | 0.0415 | 0.0502 | 0.0319 | 0.0539 | 0.0366 | 0.0389 |
| P-NARF-IC | **0.0456** | **0.0755** | 0.0490 | 0.0596 | 0.0408 | **0.0688** | 0.0450 | 0.0548 | **0.0358** | **0.0610** | **0.0439** | **0.0479** |
| improve. | 126.9% | 123.4% | 129.5% | 122.2% | 162.9% | 149.3% | 159.0% | 154.4% | 179.7% | 172.3% | 215.8% | 170.6% |

Table 3: Same setting as Table 2 except noisy level, with the experiment conducted on the real-world Task 3.

| Metric | EM CDR | PTUP CDR | E-LID | P-LID | E-NAR F-I | E-NAR F-IA | E-NAR F-IC | P-NAR F-I | P-NAR F-IA | P-NAR F-IC | improve. |
|---|---|---|---|---|---|---|---|---|---|---|---|
| R@50 | 0.0048 | 0.0052 | 0.0435 | 0.0401 | 0.0655 | 0.0840 | 0.1068 | 0.0582 | 0.0768 | **0.1268** | 216.2% |
| R@100 | 0.0091 | 0.0101 | 0.0594 | 0.0582 | 0.0921 | 0.1288 | 0.1517 | 0.0830 | 0.1279 | **0.1767** | 203.6% |
| N@50 | 0.0031 | 0.0033 | 0.0426 | 0.0345 | 0.0611 | 0.0703 | 0.0915 | 0.0493 | 0.0553 | **0.1066** | 209.0% |
| N@100 | 0.0045 | 0.0055 | 0.0481 | 0.0421 | 0.0713 | 0.0894 | 0.1105 | 0.0594 | 0.0764 | **0.1275** | 202.9% |

**Evaluation Protocol.** Following Wang et al. (2019) and He et al. (2020), the evaluation metrics recall@$k$ and ndcg@$k$ are computed as the all-ranking protocol, where all items not interacted with by a user are considered as candidates and $k = \{50, 100\}$. Refer to Appendix A.3 for additional details.

**Comparable Methods.** Base CDR algorithms are implemented with EMCDR and PTUCDR. Despite CDRIF being a relatively new problem setting, we have four baselines as follows: (i) EMCDR, (ii) PTUCDR, (iii) EMCDR with learning from implicit data (LID) (defined in Section 3) (E-LID), and (iv) PTUCDR with LID (P-LID). Similarly, six NARF-based solutions for CDRIF are: (i) EMCDR with IFC (E-NARF-I), (ii) EMCDR with IFC and AD (E-NARF-IA), (iii) EMCDR with IFC and CTD (E-NARF-IC), (iv) PTUCDR with IFC (E-NARF-I), (v) PTUCDR with IFC and AD (E–NARF-IA), and (vi) PTUCDR with IFC and CTD (E-NARF-IC).

**Experimental Results on Synthetic Tasks.** In Tables 2 and 6, we present the results of all methods, focusing on the recall@$k$ and ndcg@$k$ metrics with $k = \{50, 100\}$. These evaluations are conducted under different noise levels $\epsilon$ (10%, 15%, and 20%) for the two tasks. It is worth noting that we observe substantial improvements across all metrics when utilizing IFC, DNR, or both. In some cases, these improvements are around 200%. Particularly, both E-NARF-IC and P-NARF-IC consistently outperform other methods and achieve the highest recall and ndcg values. As we examine the recommendation performance across different levels of noise, we observe that the model's performance tends to decrease with increasing noise. This underscores the significance of denoising techniques during the learning process.

**Experimental Results on Real-world Task.** To further evaluate the performance of NARF, we conducted experiments on the real-world implicit task. The experimental results in terms of the recall@$k$ and ndcg@$k$ metrics are presented in Table 3. It can be seen that the performance of the models on this dataset is consistent with that on the Amazon review dataset. P-NARF-IC eventually can achieve more than 200% improvements. These results also highlight the effectiveness of IFC and DNR in enhancing the quality of CDRIF.

**Ablation Study.** To analyze the effectiveness of IFC and DNR, we conducted an ablation study with both. (1) **NARF w/o IFC** does not consider different calibration factors among user-item pairs. (2) **NARF w/o DNR** does not consider the negative effects of noisy data during the training. The results in Table 4 show that both the effectiveness of IFC and DNR show in general.

**How the Noise in Implicit Feedback Affects Performance.** Here, we examined the training curves of validation recall@$k$ and ndcg@$k$ for both EMCDR-based NARF and PTUPCDR-based NARF.

Table 4: Ablation study. We show the result on the real-world dataset in this table. *E* and *P* represent base CDR algorithms, EMCDR and PTUPCDR. *diff.* denotes the differences in percentage in the case of with or without *IFC* and *CTD*.

| Metric | E-w/ IFC +CTD | E-wo IFC | diff. | E-wo CTD | diff. | P-w/ IFC +CTD | P-wo IFC | diff. | P-wo CTD | diff. |
|---|---|---|---|---|---|---|---|---|---|---|
| **R@50** | 0.1068 | 0.0999 | -6.46% | 0.0655 | -38.67% | 0.1239 | 0.0744 | -39.98% | 0.0306 | -75.30% |
| **R@100** | 0.1517 | 0.1451 | -4.35% | 0.0921 | -39.29% | 0.1768 | 0.1322 | -25.23% | 0.0469 | -73.47% |
| **N@50** | 0.0915 | 0.0851 | -6.99% | 0.0611 | -33.22% | 0.1066 | 0.0524 | -50.89% | 0.0237 | -77.77% |
| **N@100** | 0.1105 | 0.1043 | -5.61% | 0.0713 | -35.48% | 0.1275 | 0.0773 | -39.34% | 0.0305 | -76.08% |

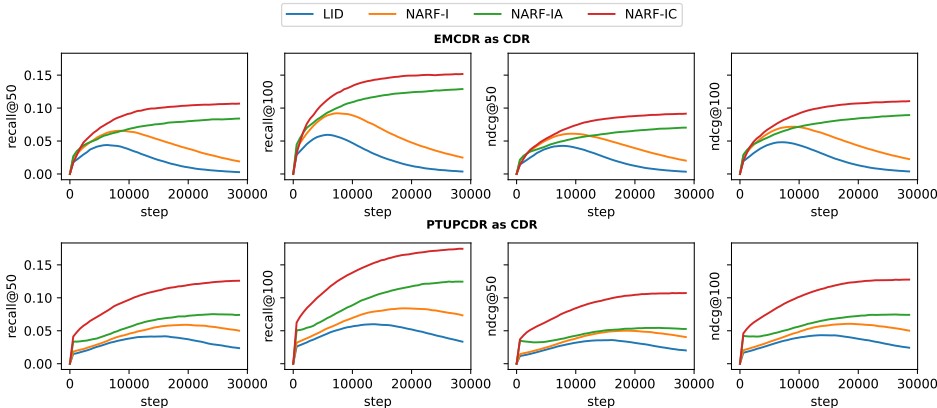

Figure 4: Illustration of how the noise in CDRIF affects performance. The first row corresponds to the methods based on EMCDR and the second row is PTUPCDR.

As depicted in Figure 4, a clear trend emerges. Both denoising methods, AD and CTD, consistently exhibit increasing performance outcomes, which is a positive indicator. In contrast, the methods without denoising start to show a decrease in performance after a few iterations, ultimately leading to unsatisfactory results. This phenomenon can be attributed to the nature of deep learning models. Initially, these models tend to memorize easy instances, which are mostly clean data since these easy instances match the distribution of the true data. Gradually, deep learning models adapt to harder instances due to their high capacity for fitting complex patterns. However, when noisy labels are present, deep learning models may eventually memorize these inaccurately provided labels, resulting in poor generalization performance. Indeed, this observation underscores the critical importance of identifying and effectively handling noisy data within implicit feedback during the training process. It is a crucial step in ensuring the robustness and generalization ability of recommendation models.

**Impact of Discarding Strategy in Denoising Methods.** In the context of DNR in CDRIF, there are three strategies for selecting denoised data. In the study by Wang et al. (2021b), only positive pairs were selected for noise reduction. However, we conducted experiments involving other cases, including only denoising negative pairs and denoising both positive and negative pairs. The results are presented in Figure 5, and they reveal an interesting pattern. We found that only denoising positive pairs resulted in the worst performance across all experimental cases, followed by only denoising negative pairs. In contrast, denoising both positive and negative pairs consistently achieved the best performance, with performance continuing to improve over time. This observation suggests that in implicit datasets, positive pairs contain more valuable information than negative pairs. The experimental results indicate that it is not appropriate to focus only on denoising the positive pairs.

# 6 CONCLUSION

This paper introduces a relatively practical problem setting known as *cross-domain recommendation from implicit feedback* (CDRIF), which addresses scenarios where explicit feedback expressing users' preferences cannot be collected. This aligns closely with real-world situations and deserves significant attention in the field of recommendation systems. To tackle this challenge, we propose a novel framework, namely *noise-aware reweighting framework* (NARF). Experiments conducted on both synthetic and large real-world datasets illustrate the effectiveness of the proposed frame-

work. Its model-agnostic and plug-and-play nature facilitates easy implementation and integration into current systems, a significant advantage over existing methods.

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

# A APPENDIX

## A.1 RELATED WORK

Early recommendation methods primarily concentrated on explicit feedback (Salakhutdinov et al., 2007; Sarwar et al., 2001; Koren et al., 2009; Koren, 2008), yet there is a discernible shift in recent research towards an increasing focus on implicit feedback data (He et al., 2016; Liang et al., 2016; Bayer et al., 2017) as collecting implicit feedback is more straightforward in real-world scenarios. Collaborative filtering (CF) tasks involving implicit feedback are frequently framed as item recommendation problems. In these scenarios, the objective is to rank items based on user preference. Unlike rating prediction, which has seen substantial progress in the context of explicit feedback, tackling item recommendation is more pragmatic yet inherently (He et al., 2015; Bayer et al., 2017). Many methods have been proposed for recommendation using implicit feedback data (Hu et al., 2008; Rendle et al., 2012; He et al., 2016; 2017; Chen et al., 2020). There are two representative learning strategies employed to mitigate the issue of lacking negative pairs when learning from implicit feedback data: *Negative Sampling* (Chen et al., 2019a; He et al., 2017; Rendle et al., 2012) and *Non-sampling*(Chen et al., 2019b; He et al., 2016; Hu et al., 2008) methods. Both strategies come with their respective strengths and weaknesses. Negative sampling, for instance, exhibits greater efficiency thanks to the restricted number of training instances. However, it may encounter performance limitations stemming from the potentially low quality of the sampled negative examples and relatively slower convergence (Chen et al., 2019b; Lian et al., 2020; Xin et al., 2018; Yuan et al., 2018). The non-sampling strategy, on the other hand, generally has the potential to attain superior performance since it fully utilizes all available training data. However, it may suffer from inefficiency as a potential drawback (Chen et al., 2020; He et al., 2016).

## A.2 DATASET AND EXPERIMENTAL SETUPS

**Dataset** In this study, we conduct experiments on a real-world public dataset, namely the **Amazon review** dataset[1] following existing cross-domain algorithms (Kang et al., 2019; Zhao et al., 2020; Zhu et al., 2021; 2022). The dataset used specifically contains ratings-only data, devoid of any metadata or text reviews, only consisting of (item, user, rating, timestamp) tuples. Out of the dataset's 24 categories, we opt to focus on three highly popular ones:: movies_and_tv (Movie), cds_and_vinyl (Music) and books (Book). For this study, we define two CDR tasks: Task 1, involving recommendations from Movie to Music, and Task 2, involving recommendations from Book to Movie. However, it is important to note that the original dataset primarily contains explicit feedback, which does not perfectly align with our problem setting. To adapt the dataset for the evaluation, several prepossessing steps are undertaken. Initially, interactions with a rating score below 3 are removed. For those interactions with scores greater than 2, the scores are adjusted by subtracting 3 to ensure that ratings start from 0. Additionally, some level of noise is introduced into the labels to simulate a more realistic scenario. Importantly, all available data is retained to mimic conditions closer to real-world applications, where data may be noisy and diverse.

To further validate our approach, we proactively collected two larger datasets from two large bibliometric datasets **PubMed**[2] and **DBLP**[3] respectively, which provide implicit feedback data. The first dataset is called PubMed, which is largely used to find references and abstracts on topics related to the life sciences and biomedicine. The second dataset is a public DBLP computer science bibliography, which contains important computer science publications and proceedings.

The DBLP and PubMed datasets are both from the Microsoft Academic Graph (MAG)[4]. Our selection of these datasets was driven by their comprehensive features, including multiple data source identifiers (DBLP and PubMed IDs, etc.), disambiguated authors, institutions and uniquely identified research topics. This rich data structure facilitated the alignment of the two datasets and the extraction of pre-processed author and topic information. The research topics are extracted via a hierarchical concept extraction tagging algorithm developed by the Microsoft research team (Shen

---

[1] https://nijianmo.github.io/amazon/index.html
[2] https://pubmed.ncbi.nlm.nih.gov/
[3] https://dblp.org/
[4] https://www.aminer.cn/oag-2-1

Table 5: Statistics of the CDRIF tasks.

| CDR Tasks | Domain | | #Item | | #User | | | #Feedback | |
|---|---|---|---|---|---|---|---|---|---|
| | Source | Target | Source | Target | Source | Target | Overlap | Source | Target |
| Task 1 | Movie | Music | 28,396 | 36,317 | 70,568 | 26,907 | 6,096 | 1,538,423 | 647,199 |
| Task 2 | Book | Movie | 82,146 | 7,135 | 387,265 | 45,786 | 11,753 | 9,607,365 | 922,763 |
| Task 3 | PubMed | DBLP | 74,990 | 33,515 | 608,308 | 106,533 | 17,370 | 11,806,446 | 1,497,925 |

et al., 2018). We defined an author's topic collection as the aggregate of topics from all papers they have authored. The specific steps for extracting this subset of data were as follows:

1. Download the MAG dataset,

2. Download the **V12 DBLP dataset**[5] and retrieve the paper IDs, which are aligned with the MAG dataset.

3. Extract the records with PubMed ID information from the MAG dataset to form the PubMed dataset (source domain) and similarly extract records for the DBLP dataset (target domain).

For our experimental analysis, we chose to focus primarily on the topic as the subject matter. This decision was based on our belief that the topic provides a more meaningful and valuable context for demonstrating the effectiveness of our algorithm. While venue and co-authorship data were considered, we determined that the topic would likely offer more significant and relevant insights for our study.

For both of these datasets, we focus on data spanning the years 2014 to 2019. Each data instance in these datasets consists of a tuple, encompassing author identification, topic identification, and the year and number of publications the author has contributed to a specific topic. This "number of publications" metric reflects an author's activity in producing works on a particular topic in a given year. The final statistics of the two datasets are listed in Table 5, For the experiment, we define this as Task 3: involving recommendations from PubMed to DBLP.

**Experimental Setups**    We employ a filtering criterion where users with fewer than 10 interactions and items with fewer than 10 interactions are filtered out. For each task, we split the historical interactions of each user into three sets: 80% for training, 10% for validation, and 10% for testing. The validation set is utilized for hyperparameter tuning, while the test set is used to assess the model's generalization performance. To evaluate the model's performance in cold-start scenarios, we deliberately remove all rating information for 20% of the users in the target domain, treating them as cross-domain cold-start users for the purpose of making recommendations. For tasks 1 and 2, which simulate implicit feedback scenarios, we introduce different noise levels of 10%, 15%, and 20%, denoted as $\epsilon$, to assess the effectiveness of CDR.

## A.3   EVALUATION PROTOCOL

Following evaluation protocol of recent recommendation task (Wang et al., 2019; He et al., 2020). For each dataset, we randomly allocate 80% of a user's historical interactions to form the training set, while the remainder is designated as the test set. Additionally, from the training set, we randomly extract 10% of interactions to create a validation set, which is used for hyperparameter tuning. To assess the efficacy of top-K recommendation and preference ranking, we employ two commonly used evaluation protocols, recall@$k$ and ndcg@$k$ (He et al., 2017; Yang et al., 2018). In this study, we configure K to be 50 and 100. Our reported metrics represent the average performance across all users in the test set.

## A.4   IMPLEMENTATION DETAILS

Our framework is implemented using PyTorch(Paszke et al., 2019) and PyTorch Lightning (Falcon & The PyTorch Lightning team, 2019). To optimize the performance, we utilize the validation set for

---

[5]`https://www.aminer.org/citation`

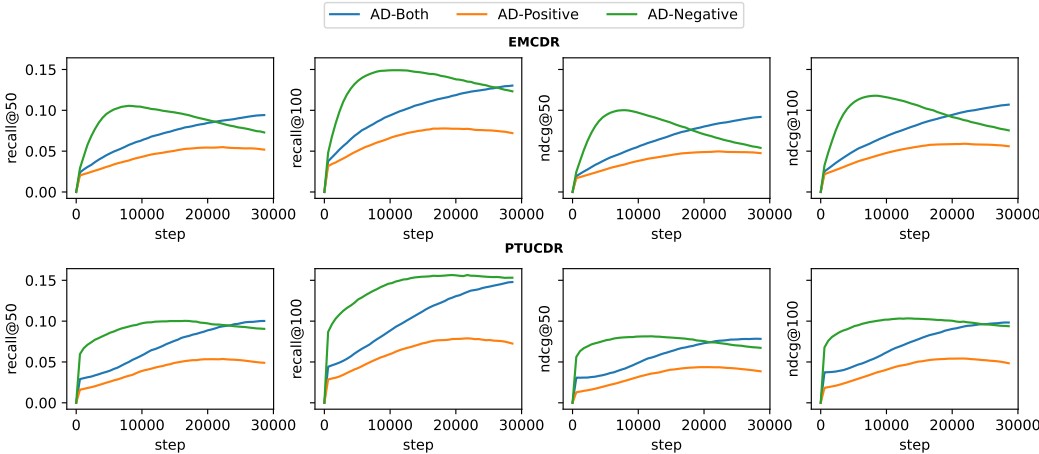

Figure 5: Training curves between three discarding strategies for the Adaptive Denoising (AD).

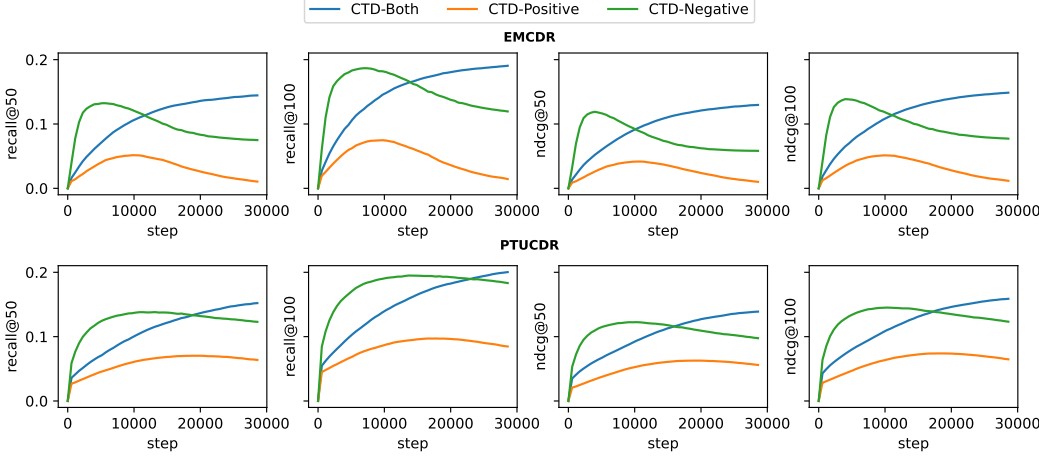

Figure 6: Training curves between three discarding strategies for the Co-teaching Denoising (CTD).

hyperparameter tuning. The default optimizer is Adam (Kingma & Ba, 2014), with an initial learning rate of 0.001 and a batch size of 1024. To mitigate overfitting, we incorporate L2 regularization with a value of 0.00001. In addition, we set the dimension $d$ of embeddings of each method at 128 and the mapping dimension to 64. Regarding the denoising strategies, we consider the following hyperparameters: the forget rate $tau$, the exponent $c$, and the gradual number $T_k$. Specifically, $tau$ is explored within the range $\{0.01, 0.05, 0.1, 0.2\}$, $c$ is tuned using values from $\{0.5, 1.0, 2.0\}$, and we experiment with three different values for $T_k$, namely $T_k = \{5, 10, 15\}$.

Table 6: Same setting as Table 2, with the experiment conducted for Task 2.

| Method | $\epsilon = 10\%$ | | | | $\epsilon = 15\%$ | | | | $\epsilon = 20\%$ | | | |
|---|---|---|---|---|---|---|---|---|---|---|---|---|
| | R@50 | R@100 | N@50 | N@100 | R@50 | R@100 | N@50 | N@100 | R@50 | R@100 | N@50 | N@100 |
| EMCDR | 0.0132 | 0.0202 | 0.0112 | 0.0131 | 0.0122 | 0.0162 | 0.0092 | 0.0111 | 0.0092 | 0.0101 | 0.0075 | 0.0098 |
| PTUPCDR | 0.0140 | 0.0238 | 0.0129 | 0.0169 | 0.0123 | 0.0175 | 0.0112 | 0.0148 | 0.0106 | 0.0116 | 0.0087 | 0.0105 |
| E-LID | 0.0237 | 0.0410 | 0.0200 | 0.0270 | 0.0235 | 0.0398 | 0.0209 | 0.0274 | 0.0183 | 0.0320 | 0.0171 | 0.0223 |
| P-LID | 0.0240 | 0.0401 | 0.0214 | 0.0278 | 0.0210 | 0.0367 | 0.0189 | 0.0252 | 0.0193 | 0.0330 | 0.0176 | 0.0231 |
| E-NARF-I | 0.0349 | 0.0610 | 0.0304 | 0.0408 | 0.0321 | 0.0562 | 0.0271 | 0.0370 | 0.0297 | 0.0511 | 0.0259 | 0.0346 |
| E-NARF-IA | 0.0496 | 0.0847 | 0.0435 | 0.0574 | 0.0454 | 0.0764 | 0.0408 | 0.0531 | 0.0415 | 0.0705 | 0.0363 | 0.0477 |
| E-NARF-IC | **0.0553** | **0.0919** | **0.0480** | **0.0623** | 0.0492 | 0.0816 | 0.0441 | 0.0569 | **0.0479** | **0.0776** | **0.0414** | **0.0531** |
| P-NARF-I | 0.0354 | 0.0609 | 0.0310 | 0.0413 | 0.0293 | 0.0518 | 0.0254 | 0.0345 | 0.0283 | 0.0507 | 0.0252 | 0.0342 |
| P-NARF-IA | 0.0491 | 0.0842 | 0.0428 | 0.0569 | 0.0418 | 0.0720 | 0.0359 | 0.0468 | 0.0394 | 0.0649 | 0.0339 | 0.0441 |
| P-NARF-IC | 0.0542 | 0.0893 | 0.0469 | 0.0607 | **0.0496** | **0.0824** | **0.0442** | **0.0570** | 0.0437 | 0.0730 | 0.0391 | 0.0505 |
| improve. | 130.4% | 124.1% | 124.3% | 124.1% | 111.1% | 107.0% | 111.5% | 126.2% | 148.2% | 135.2% | 135.2% | 129.9% |

