# OpenReview forum: "Cross-domain Recommendation from Implicit Feedback"
_ICLR.cc/2024/Conference — Submitted to ICLR 2024_

### Official Review · Reviewer_VYJS · 2023-10-30

**Soundness:** 2 fair
**Presentation:** 3 good
**Contribution:** 2 fair
**Rating:** 5
**Confidence:** 5

**Summary:**

The paper introduces Neural Adaptive Recommendation Feedback (NARF), a novel approach designed to enhance Cross-Domain Recommendation with Implicit Feedback (CDRIF). NARF comprises two main components: Implicit Feedback Calibration (IFC) and Dynamic Noise Reduction (DNR), aiming to address the challenges posed by the binary nature and noise in implicit feedback. The authors conduct extensive experiments on both synthetic and real-world datasets, demonstrating that NARF significantly outperforms baseline methods. The paper is well-structured, providing clear explanations of the methodology and a comprehensive evaluation of the proposed approach.

**Strengths:**

1.	The introduction of NARF as a method to handle implicit feedback in CDR is innovative. The two-step approach of IFC and DNR is well thought out and addresses the challenges of implicit feedback effectively.
2.	The paper includes a comprehensive set of experiments on both synthetic and real-world datasets, providing a thorough evaluation of the proposed method. The results show that NARF significantly outperforms baseline methods.
4.	The paper provides a clear and detailed description of the NARF approach, including the mathematical formulations and the rationale behind each component.

**Weaknesses:**

1.  It is important to balance denoising and preservation of unique user vehaviors : This weakness pertains to the treatment of implicit feedback in the denoising strategy and the Dynamic Noise Reduction (DNR) process within the NARF framework. The concern is that these processes might overly standardize user behavior, potentially leading to the loss of unique and valuable user behavior patterns inherent in implicit feedback.

 n the NARF framework, implicit feedback is subjected to a denoising strategy to mitigate the impact of noisy interactions. Implicit feedback, by nature, is derived from user actions such as clicks, views, or purchases, and it encapsulates a wide array of user behavior patterns. These patterns are crucial as they provide unique insights into user preferences and behavior. However, the denoising strategy, as described in the paper, might not sufficiently differentiate between noise and genuine user behavior patterns that are less common or more nuanced. This is particularly evident in the sections of the paper where the authors discuss the implementation of the DNR process.

The DNR process aims to dynamically reduce noise by focusing on reliable user-item interactions. However, the criteria for determining reliability are primarily based on loss values computed during training. This approach raises the concern that unique user behavior patterns, which might initially result in higher loss values due to their rarity or complexity, could be mistakenly treated as noise and filtered out. This is especially problematic if these unique patterns are crucial for understanding specific user preferences or behavior in niche domains.

Furthermore, the paper does not provide a detailed discussion on how the NARF framework, and specifically the DNR process, handles the trade-off between denoising and preserving the richness of implicit feedback. The lack of this discussion leaves readers questioning whether the framework can adequately capture the diversity of user behavior inherent in implicit feedback, particularly in scenarios with diverse and biased datasets.

In summary, while the NARF framework aims to improve recommendation performance through denoising strategies, there is a need for a more explicit discussion and concrete examples in the paper to address concerns regarding the potential loss of inherent user behavior patterns in the implicit preferences. This is crucial for ensuring that the benefits of denoising do not come at the expense of overlooking valuable insights that implicit feedback can provide.

2. The cross-domain datasets used in the experiments are all sourced from Amazon, covering domains like movies, music, and books. These domains are relatively similar, and the use of more diverse and biased datasets could provide a more comprehensive evaluation of the proposed method.

3. It is imperative to maintain a consistent use of terminology when expressing the substitution of DNR with CTD in the ablation study, ensuring clarity and precision in communication. For instance, in Table 4, the term 'CTD' is utilized, whereas in the main model description sections, such as in Figure 3, the term 'DNR' is employed. This discrepancy necessitates attention to ensure consistency and avoid potential confusion for readers.

**Questions:**

•	How does the NARF framework ensure the preservation of unique user behavior patterns during the denoising process of implicit feedback? Are there specific mechanisms or safeguards in place to prevent the loss of these valuable insights?

•	"In the context of denoising implicit feedback, how does NARF balance the need for standardizing user behavior with the necessity to retain the inherent uniqueness of user interactions? Could you provide examples or scenarios where this balance is particularly crucial?

•	Could you elaborate on the strategies employed within the DNR process to avoid over-denoising? How does the system determine the optimal level of noise reduction to ensure that significant user behavior patterns are not inadvertently filtered out?

•	How might the performance of NARF vary when applied to more diverse and biased cross-domain datasets?

---

> ### Author Response · Authors · 2023-11-20
>
> Dear Reviewer VYJS,
>
> Thank you for acknowledging the innovative aspects of our NARF framework and its comprehensive evaluation. We appreciate your detailed feedback and would like to address your concerns regarding the denoising strategy and the datasets used in our experiments.
>
> > Q1: How does the NARF framework ensure the preservation of unique user behavior patterns during the denoising process of implicit feedback? Are there specific mechanisms or safeguards in place to prevent the loss of these valuable insights?
>
> **Response 1:** It's important to note that in practical scenarios, we often lack precise information about the amount of noise present in a dataset. Our approach, therefore, relies on using the validation set to determine the appropriate level of denoising. This process involves deciding how many reliable user patterns should be retained during training to ensure the integrity of the data.
>
> Based on this approach, we then evaluate the performance of our method on the test set. In our experiments, we observed a significant improvement when comparing our denosing method against approaches that do not incorporate such a mechanism. This marked enhancement in performance, as evidenced by our empirical results, strongly indicates the effectiveness of our method.
>
> By focusing on retaining useful and reliable user patterns while eliminating noise, our approach demonstrates its capability to enhance the quality and reliability of the data used in the training process, leading to more accurate and robust outcomes.
>
> > Q2: In the context of denoising implicit feedback, how does NARF balance the need for standardizing user behavior with the necessity to retain the inherent uniqueness of user interactions? Could you provide examples or scenarios where this balance is particularly crucial?
>
> > Q3: Could you elaborate on the strategies employed within the DNR process to avoid over-denoising? How does the system determine the optimal level of noise reduction to ensure that significant user behavior patterns are not inadvertently filtered out?
>
> **Response 2:** We appreciate your insightful concern regarding the balance between denoising and preserving unique user behaviors in implicit feedback scenarios. Let us clarify our approach:
>
> Our framework's primary objective is to enhance recommendation performance by addressing the inherent noisiness of implicit datasets. This noise often leads to suboptimal recommendation results. Through our experiments, we have demonstrated that effectively removing this noise significantly boosts performance.
>
> Addressing your query about maintaining this delicate balance, our strategy involves utilizing the validation set to fine-tune the denoising factor. This factor is crucial as it dictates the equilibrium between eliminating noise and retaining the richness and diversity of implicit feedback. The optimal denoising factor identified through the validation set is expected to yield favorable results on the test set. Our experimental findings affirm the validity of this approach.
>
> However, upon reflection and considering your suggestion, I realize that our paper could benefit from a more detailed discussion on the workings of the NARF framework, particularly in the context of this balance. We acknowledge this as a way to improve the readability of our paper.
>
> > Q4: How might the performance of NARF vary when applied to more diverse and biased cross-domain datasets?
>
> **Response 3:** **There might be a misunderstanding.** In our original paper, we not only utilized the Amazon synthetic dataset but also collected a substantial real-world dataset, referred to as Task 3. This dataset is particularly important as it reflects the complexities and challenges encountered in real-world scenarios, especially where the proportion of noisy data is difficult to ascertain. The motivation for collecting this dataset stems from a critical observation: existing Cross-Domain Recommendation from Implicit Feedback (CDRIF) algorithms predominantly rely on synthetic data. We identified a potential bias in results derived solely from such data. To address this, we gathered real-world data and applied our method to establish a more realistic and robust benchmark. We will later open source this dataset for scientific research purposes to encourage more work in this field.
>
> > It is imperative to maintain a consistent use of terminology when expressing the substitution of DNR with CTD in the ablation study, ensuring clarity and precision in communication.
>
> **Response 4:** We appreciate your pointing out the inconsistency in terminology between 'CTD' and 'DNR'. We will ensure that the terms are used consistently to avoid confusion.
>
> **Overall**, we thank you for your constructive feedback, which will greatly help in improving our work. We do hope our responses can address your concerns, and **your re-evaluation is very important for us (for this time or future).**

---

> > ### Author Response · Authors · 2023-11-22
> >
> > Dear Reviewer VYJS,
> >
> > I hope this message finds you well. I am writing to provide an update on the additional experiments we have conducted in response to the feedback. As suggested, we have included comparative results with two additional baselines: DCDCSR (Zhu, Feng, et al. A deep framework for cross-domain and cross-system recommendations. In IJCAI 2018.) and LTR (Chen, Jin, et al. Learning Recommenders for Implicit Feedback with Importance Resampling. Proceedings of the ACM Web Conference 2022.). As LTR is not the algorithm for CDR, we incorporate it with DCDCSR as DCDCSR-LTR.
> >
> > The results of these experiments are as follows:
> >
> > | **Metric** | **DCDCSR** | **DCDCSR-LTR** | **Ours** |
> > | :-: | :-: | :-: | :-: |
> > | **R@50** | 0.0058 | 0.0575 | **0.1268** |
> > | **R@100** | 0.0098 | 0.0721 | **0.1767** |
> > | **N@50** | 0.0042 | 0.0512 | **0.1066** |
> > | **N@100** | 0.0053 | 0.0589 | **0.1275** |
> >
> > These results demonstrate the significant improvement our method offers over the aforementioned baselines. We believe this additional evidence further strengthens our method. We hope this addresses any concerns and provides a clearer understanding of our work's contribution. In this regard, it would be greatly appreciated if you could take a moment to provide any further feedback or comments on our rebuttal response. We understand the time and effort involved in this process and are grateful for your continued guidance and expertise.
> >
> > Thank you once again for your time and consideration. We look forward to any further thoughts you might have.

---

### Official Review · Reviewer_Bnev · 2023-11-02

**Soundness:** 1 poor
**Presentation:** 1 poor
**Contribution:** 1 poor
**Rating:** 3
**Confidence:** 4

**Summary:**

The paper introduces a cross domain recommendation methods based on denoising and calibrating the user preferences in the source and target domain. The experimental results seem promising but the paper is essentially unreadable with major issues in presentation. In particular the method is extremely difficult to understand, e.g. how are the calibration factors k_i computed from function 14 useful in determining the true preferences of the user, no intuition is provided. Moreover the paper mainly deals with matrix factorization type methods and there is not much discussion about more modern deep learning type methods and how it relates to those. Moreover the literature on cross-domain recommendation is quite big so some more extensive set of baselines could have been used.
Overall I do not see the major contribution of this work to a general ML conference like iclr and this might be more suitable for a recommender systems conference,

**Strengths:**

Seems to have a good performance compared to the used baselines.

Code availability

**Weaknesses:**

The paper is difficult to follow and does not provide good intuition on why the design choices where made.

Compared to the literature on the topic the authors use a very limited set of baselines.

The topic is a rather niche topic in the area of recommender systems and might not be interesting to the wider audience at iclr.

**Questions:**

Could the authors provide more intuition on the design choices of the algorithm.
I also find papers with too many abbreviations very hard to follow.
Simplify the presentation of the paper.

---

> ### Author Response · Authors · 2023-11-20
>
> Dear Reviewer Bnev,
>
> Thank you for your feedback and for acknowledging the performance of our work compared to the selected baselines. We appreciate your insights and would like to address the concerns raised.
>
> > The paper is difficult to follow and does not provide good intuition on why the design choices were made.
>
> **Response 1:** We understand your concerns about the clarity and presentation of the paper. In the revised version, we will:
> - **Simplify the Presentation:** Reduce the use of abbreviations and present the concepts in a more straightforward manner.
> - **Provide Intuition for Design Choices:** We introduced our ideas by explaining the current problems with CDRIF and the challenges that implicit feedback naturally presents in the introductory section. Let us know if you have any specific questions.
>
> > Compared to the literature on the topic the authors use a very limited set of baselines.
>
> **Response 2:** We understand your concern regarding the baseline methods selection. In our study, we focused on methods most relevant to our specific research question. However, we acknowledge the omission of certain important works in implicit feedback recommendation systems. We agree that including these methods could provide a more comprehensive understanding. To address this, we are actively expanding our experimental scope to include these important methods. We are committed to ensuring a thorough and robust evaluation of our framework.
>
> **We will promptly report the updated results once these additional experiments are completed**.
>
> > The topic is a rather niche topic in the area of recommender systems and might not be interesting to the wider audience at iclr.
>
> **Response 3:** Although there are many papers focusing on common cross-domain recommendations, they are limited in explicit feedback scenarios. From this point of view, our proposed framework is more general and wider in real-world applications. Thus, in our humble opinion, **we cannot agree** that our topic is a rather niche topic in the area of recommender systems. **It is not reasonable to deny a research direction just because the amount of related literature is small.**
>
> **Recommendation is an important research topic.** Recommendation is an important problem in the real world and can be regarded as a classification problem but with special features (normally user/item ids). **Recommendation methods are very useful in the real world and have been successfully deployed in many daily-used websites or systems**. The significance of to study recommendation is clear.
>
> **Our contributions to the field.** Our proposed framework addresses a significant challenge in recommendation, making effective recommendations in domains with implicit feedback. We believe this has substantial relevance given the prevalence of such scenarios in real-world applications. Also as a model-agnostic and plug-and-play approach, it is easy to implement and deploy to the existing systems in practice.
>
>
> **In conclusion**, we thank you for your constructive feedback. We are committed to improving our work based on your suggestions and hope our revisions will address your concerns adequately. Finally, we will update a new version of our paper here (we are running some experiments now and will try our best to update our paper in the end of the author-reviewer discussion). We do hope to get your new responses for our updated paper, and **your re-evaluation is very important for us (for this time or future).**

---

> > ### Author Response · Authors · 2023-11-22
> >
> > Dear Reviewer Bnev,
> >
> > I hope this message finds you well. I am writing to provide an update on the additional experiments we have conducted in response to the feedback. As suggested, we have included comparative results with two additional baselines: DCDCSR (Zhu, Feng, et al. A deep framework for cross-domain and cross-system recommendations. In IJCAI 2018.) and LTR (Chen, Jin, et al. Learning Recommenders for Implicit Feedback with Importance Resampling. Proceedings of the ACM Web Conference 2022.). As LTR is not the algorithm for CDR, we incorporate it with DCDCSR as DCDCSR-LTR.
> >
> > The results of these experiments are as follows:
> >
> > | **Metric** | **DCDCSR** | **DCDCSR-LTR** | **Ours** |
> > |:----------:|:----------:|:---------:|:----------:|
> > | **R@50**   | 0.0058     | 0.0575    | **0.1268** |
> > | **R@100**  | 0.0098     | 0.0721    | **0.1767** |
> > | **N@50**   | 0.0042     | 0.0512    | **0.1066** |
> > | **N@100**  | 0.0053     | 0.0589    | **0.1275** |
> >
> > These results demonstrate the significant improvement our method offers over the aforementioned baselines. We believe this additional evidence further strengthens our method. We hope this addresses any concerns and provides a clearer understanding of our work's contribution. In this regard, it would be greatly appreciated if you could take a moment to provide any further feedback or comments on our rebuttal response. We understand the time and effort involved in this process and are grateful for your continued guidance and expertise.
> >
> > Thank you once again for your time and consideration. We look forward to any further thoughts you might have.

---

### Official Review · Reviewer_BpkS · 2023-11-03

**Soundness:** 2 fair
**Presentation:** 3 good
**Contribution:** 1 poor
**Rating:** 1
**Confidence:** 5

**Summary:**

This paper investigates the implicit feedback-based (e.g., click, purchase, count of behaviors, in contrast to the explicit feedback e.g., ratings with positive and negative preferences) recommendation problem in the cross-domain setting. The two challenges in this setting are the absence of negative signals and the noise of user-item interactions. The proposed framework (CDRIF) addresses the first issue by existing sampling strategies, i.e., uniform sampling (Rendle et al., 2012; Ding et al., 2020). And CDRIF addresses the second issue by existing denoising algorithms, i.e., adaptive denoising and co-teaching denoising (Wang et al., 2021b; Han et al., 2018). CDRIF introduces an item popularity-based calibration factor as the weight of loss function. CDRIF shows improvement over two baselines (EMCDR and PTUPCDR) on two datasets.

**Strengths:**

S1: Using a popularity-based calibration factor to weigh the loss function seems promising as an alternative to attention-based methods.

S2: The writing is easy to follow.

**Weaknesses:**

W1: The technical contributions are very limited. The proposed framework (CDRIF) consists of three core components, i) generating negative samples, ii) reducing noise, iii) reweighting by calibration factors. However, the first component directly uses the existing uniform sampling (Rendle et al., 2012; Ding et al., 2020). The second component directly uses existing denoising algorithms, i.e., adaptive denoising and co-teaching denoising (Wang et al., 2021b; Han et al., 2018). The third component is a very simple item popularity-based calibration.

W2: The evaluation is very weak. There are only two baselines, EMCDR (Man et al., 2017) and PTUPCDR (Zhu et al., 2022). For cross-domain recommendation (CDR) methods (applicable for both explicit and implicit feedback), there are so many advanced learning techniques, just name a few bellows: graph neural networks based CDR, transfer learning based CDR, attention mechanism based CDR, adversarial learning based CDR.

W3: Many related works are ignored. Also, see the above weak W2. In detail, transfer learning is a main research thread to address the cross-domain recommendation (for both explicit and implicit feedback). For example, the CoNet method (Collaborative Cross Networks for Cross-Domain Recommendation) can address the cross-domain recommendation from implicit feedback as investigated in this ICLR submission. In its task setting on the Cheetah Mobile dataset, it recommends apps (the target domain) by exploiting knowledge from news reading history (the source domain). Obviously, the reading logs are implicit feedback and the installations of apps are also implicit feedback. As a result, the claim in the introduction “to the best of our knowledge, prior to our study, no existing CDR research has offered algorithms to handle implicit feedback scenarios” does not hold true.

W4: the newly introduced dataset is not detailed. I read the Appendix A.2 for the dataset description on PubMed and DBLP. I checked the web pages on PubMed and DBLP, but I still do not know: How to align the authors on these two domains (name disambiguation)? Where to get the topic identification of an author? Why use the topic as the items and what is the reason to recommend a topic to an author? Why not use the {venue, paper/references, potential co-author et al} as the items instead? Is any data sample to be shown? By the way, will this dataset be released to use for research purposes? What is the cleaning, filtering, and preprocessing detail for this newly introduced dataset?

**Questions:**

Q1: The PTUPCDR refers to (Zhu et al., 2020) (A deep framework for cross-domain and cross-system recommendations) in the Section "Realization of NARF: DNR", while PTUPCDR refers to (Zhu et al., 2022) (Personalized transfer of user preferences for cross-domain recommendation) in Table 1.

Q2: how to determine the two hyperparameters alpha and beta in Eq. (14)?

Q3: In Eq. (14), for the sigma summation, $m$ should be $n$ which denotes the number of items instead of the number of users.

Q4: Below Eq. (14), what is $f_j$? And what is the relation between $f_j$ and $p_j$

Q5: In Eq. (14), for the positive pairs, the control hyperparameter alpha is the same for all user-item pairs. How about learning such control hyperparameter alpha_{i,j} according to individual user-item (u_i, v_j) pairs? This may be achieved by attention-based networks.

Q6: checking the syntax issue for this sentence, “the evaluation metrics
are recall@k and ndcg@k are computed following the all-ranking protocol”, which has two “are”s.

Q7: In Tables 2 and 6, how about the noise level equals zero?

Q8: there are so many implicit feedback benchmarks like Criteo (CTR clicks) and Microsoft MIND (news clicking). Why not ignore evaluations on such large, real-world datasets? By the way, the evaluated Amazon review dataset is NOT an implicit feedback dataset.

---

> ### Author Response · Authors · 2023-11-20
> **Reply - Part I**
>
> Dear Reviewer BpkS,
>
> Thank you for your thorough review and insightful comments. We appreciate the opportunity to clarify and address the concerns raised.
>
> > The technical contributions are very limited.
>
> **Response 1:** Our work specifically addresses Cross-Domain Recommendation from Implicit Feedback (CDRIF). We introduce an innovative framework tailored to this particular issue. This framework, designed to be compatible with existing techniques, has demonstrated its effectiveness through rigorous experiments. Its model-agnostic and plug-and-play nature facilitates easy implementation and integration into current systems, a significant advantage over existing methods.
>
> Addressing the critique of our technical contribution, it is important to note the distinctiveness of our framework. Developing a versatile framework compatible with a range of existing techniques, especially for CDRIF, is a non-trivial task. Our work pioneers in this area, marking a significant technical advancement. **Therefore, dismissing our technical contribution in designing this innovative framework from scratch would be unjust**.
>
> Our framework's uniqueness is further highlighted by its ability to address two critical challenges in recommendation systems:
>
> 1. **Cold-Start Problem in Implicit-Feedback Systems**: Traditional implicit-feedback systems struggle with this issue. Our framework, as evidenced in our experiments, effectively leverages knowledge transfer from a source domain to mitigate the cold-start problem in the target domain.
>
> 2. **Handling Implicit Feedback in CDR Methods**: Current CDR methods inadequately address implicit feedback, often resorting to a combination of traditional CDR methods and negative sampling. Our framework, however, goes beyond this by systematically investigating cross-domain recommendations under implicit feedback scenarios and proposing a comprehensive solution rather than just methodological adjustments.
>
> In summary, our paper is the **first** to thoroughly explore CDR under implicit feedback, proposing a robust and adaptable framework for CDRIF. This framework not only outperforms existing baselines with **a large margin** but also offers ease of implementation and deployment in practical settings. **Therefore, again, dismissing our contribution to designing a novel, general framework from scratch would overlook the substantial advancements our work presents in this field**.
>
> > The evaluation is very weak.
>
> **Response 2:** We understand your concern regarding the baseline methods selection. In our study, we focused on methods most relevant to our specific research question. However, we acknowledge the omission of certain important works in implicit feedback recommendation systems. We agree that including these methods could provide a more comprehensive understanding. To address this, we are actively expanding our experimental scope to include these important methods. We are committed to ensuring a thorough and robust evaluation of our framework.
>
> **We will promptly report the updated results once these additional experiments are completed**.
>
> > Many related works are ignored.
>
> **Response 3:** We appreciate your pointing out the omission of relevant works, such as the CoNet method. We will revise these and other relevant studies, ensuring a comprehensive overview of the state-of-the-art in CDR, particularly focusing on implicit feedback scenarios.
>
> > The newly introduced dataset is not detailed.
>
> **Response 4:** Thank you for highlighting the need for more detailed information about our newly introduced dataset. **This is a new dataset collected by ourselves.** The DBLP dataset and PubMed dataset are both from the Microsoft Academic Graph (MAG), which can be downloaded from this link: https://www.aminer.cn/oag-2-1. We selected this dataset as it provides multiple data source identifiers (DBLP and PubMed IDs, etc.), disambiguated authors, institutions and research topics (with unique identifiers). That explains how we aligned the two datasets and obtained the pre-processed author and topic information. The research topics are extracted via a hierarchical concept extraction tagging algorithm developed by the Microsoft research team (Shen, Z., Ma, H., & Wang, K. (2018). A web-scale system for scientific knowledge exploration. arXiv preprint arXiv:1805.12216.). we assume that an author's topic collection is the union of all topics from papers he/she has authored. The specific procedures for extracting this subset are:
> 1. Download the MAG dataset,
> 2. Download the V12 DBLP dataset (https://www.aminer.org/citation) and retrieve the paper IDs (which are already aligned with the MAG dataset).
> 3. Fetch the records with PubMed ID information from the MAG dataset as the PubMed dataset (source domain), then fetch the records from the MAG dataset as the DBLP dataset (target domain).

---

> > ### Author Response · Authors · 2023-11-20
> > **Reply - Part II**
> >
> > In our paper, we primarily recommend using the topic as the subject for our experimental analysis. This choice was made after careful consideration, as we believe that **topic** offers a more significant and valuable context for demonstrating the effectiveness of our algorithm. While venue and co-author are also viable options, we opted for the topic based on its potential to yield more impactful and relevant insights in our study. We also plan to release the dataset for research purposes, subject to obtaining the necessary permissions.
> >
> > > Q1: The PTUPCDR refers to (Zhu et al., 2020) (A deep framework for cross-domain and cross-system recommendations) in the Section "Realization of NARF: DNR", while PTUPCDR refers to (Zhu et al., 2022) (Personalized transfer of user preferences for cross-domain recommendation) in Table 1.
> >
> > **Response 5:** We apologize for the inconsistency in referencing PTUPCDR and will correct this.
> >
> > > Q2: How to determine the two hyperparameters alpha and beta in Eq. (14)?
> >
> > **Response 6:** Alpha and beta in Eq. (14) are determined through a combination of grid search and empirical evaluation on a validation set. We will add a subsection detailing this process in the updated paper.
> >
> > > Q3: In Eq. (14), for the sigma summation, *m* should be *n* which denotes the number of items instead of the number of users.
> >
> > **Response 7:** You are correct; this is a typographical error, and we will correct it.
> >
> > > Q4: Below Eq. (14), what is $f_j$? And what is the relation between $f_j$ and $p_j$?
> >
> > **Response 8:** You are correct; this is a typographical error, and we will correct it.
> >
> > > Q5: In Eq. (14), for the positive pairs, the control hyperparameter alpha is the same for all user-item pairs. How about learning such control hyperparameter alpha\_{i,j} according to individual user-item (u\_i, v\_j) pairs? This may be achieved by attention-based networks.
> >
> > **Response 9:** The suggestion to use attention-based networks for learning the hyperparameter alpha_{i,j} is insightful and could be an interesting direction for future work.
> >
> > > Q6: checking the syntax issue for this sentence, “the evaluation metrics are recall@k and ndcg@k are computed following the all-ranking protocol”, which has two “are”s.
> >
> > **Response 10:** We appreciate your pointing out the syntax error and will rectify it.
> >
> > > Q7: In Tables 2 and 6, how about the noise level equals zero?
> >
> > **Response 11:** We will include results for the case where the noise level equals zero in the updated paper and report the results here when done.
> >
> > > Q8: there are so many implicit feedback benchmarks like Criteo (CTR clicks) and Microsoft MIND (news clicking). Why not ignore evaluations on such large, real-world datasets? By the way, the evaluated Amazon review dataset is NOT an implicit feedback dataset.
> >
> > **Response 12:** We **know** that there are so many implicit feedback benchmarks, but it is not easy to find implicit feedback datasets for cross-domain scenarios. In fact, our paper has introduced a dataset we collected ourselves as the experiment results shown in Table 2.
> >
> > **In conclusion**, we thank you for your constructive feedback. We are committed to improving our work based on your suggestions and hope our revisions will address your concerns adequately. Finally, we will update a new version of our paper here (we are running some experiments now and will try our best to update our paper at the end of the author-reviewer discussion). We do hope to get your new responses for our updated paper, and **your re-evaluation is very important for us (for this time or future).**

---

> > > ### Author Response · Authors · 2023-11-22
> > >
> > > Dear Reviewer BpkS,
> > >
> > > I hope this message finds you well. I am writing to provide an update on the additional experiments we have conducted in response to the feedback. As suggested, we have included comparative results with two additional baselines: DCDCSR (Zhu, Feng, et al. A deep framework for cross-domain and cross-system recommendations. In IJCAI 2018.) and LTR (Chen, Jin, et al. Learning Recommenders for Implicit Feedback with Importance Resampling. Proceedings of the ACM Web Conference 2022.). As LTR is not the algorithm for CDR, we incorporate it with DCDCSR as DCDCSR-LTR.
> > >
> > > The results of these experiments are as follows:
> > >
> > > | **Metric** | **DCDCSR** | **DCDCSR-LTR** | **Ours** |
> > > |:----------:|:----------:|:---------:|:----------:|
> > > | **R@50**   | 0.0058     | 0.0575    | **0.1268** |
> > > | **R@100**  | 0.0098     | 0.0721    | **0.1767** |
> > > | **N@50**   | 0.0042     | 0.0512    | **0.1066** |
> > > | **N@100**  | 0.0053     | 0.0589    | **0.1275** |
> > >
> > > These results demonstrate the significant improvement our method offers over the aforementioned baselines. We believe this additional evidence further strengthens our method. We hope this addresses any concerns and provides a clearer understanding of our work's contribution. In this regard, it would be greatly appreciated if you could take a moment to provide any further feedback or comments on our rebuttal response. We understand the time and effort involved in this process and are grateful for your continued guidance and expertise.
> > >
> > > Thank you once again for your time and consideration. We look forward to any further thoughts you might have.

---

### Official Review · Reviewer_LrYi · 2023-11-03

**Soundness:** 3 good
**Presentation:** 2 fair
**Contribution:** 1 poor
**Rating:** 3
**Confidence:** 4

**Summary:**

This work focuses on tackling cross-domain recommendation problem under implicit feedback setup. To reduce the noisy signals from implicit feedback, the authors proposed a noise-aware re-weighting framework by leveraging dynamic sampling methods to optimize the embedding learning. Experimental results demonstrate the effectiveness.

**Strengths:**

1. The problem is well described and motivated.
2. The proposed solution is technically sound.
3. Experiments are well designed and the presented results demonstrate the improvement.

**Weaknesses:**

1. Technical contribution is limited. Both implicit feedback recommendation and cross-domain recommendation are well studied topics, and the proposed denoising framework is also stacked by well studied methods, e.g., negative sampling from learning to rank, AD/CTD. Moreover, the authors claim this is the first work on cross-domain recommendation with implicit feedback, however, it's not convincing. For example, "Cross-domain Recommendation Without Sharing User Relevant Data" and "User-specific Adaptive Fine-tuning for Cross-domain Recommendations" are both proposed to work on the implicit feedback data.
2. Baseline methods are not well selected. There are numbers of sampling related research in learning to rank and applied to implicit feedback recommendation methods, however, few are included in comparison or discussions, e.g., Learning Recommenders for Implicit Feedback with Importance Resampling, etc.
3. The presentation of this work needs improvement. In particular, the notations used in this paper are hard to follow and can be simplified. Moreover, the framework section is hard to follow. It's unclear why the calibration factor can act as a denoising factor. What's the difference between this proposed idea and other active learning ideas?

**Questions:**

1. What's the main technical contribution for this work if this is not the first one tackling cross-domain recommendation with implicit feedback?
2. Why the learnable calibration parameter can help denoising? What's the difference between this idea and the active learning idea?
3. What's the performance comparison with state-of-the-art negative sampling method in learning to rank?

---

> ### Author Response · Authors · 2023-11-20
> **Reply - Part I**
>
> Dear Reviewer LrYi,
>
> Thank you for your valuable feedback and for highlighting areas where our submission can be improved. We appreciate the opportunity to address your concerns.
>
> > Technical contribution is limited.
>
> > What's the main technical contribution for this work if this is not the first one tackling cross-domain recommendation with implicit feedback?
>
> **Response 1:** We appreciate your concerns about the novelty of our work. However, we assert that our contribution is notably distinct from the studies you have mentioned. The referenced research primarily focuses on Cross-Domain Recommender (CDR) algorithm design and primarily **involves experiments with implicit datasets**. However, their approach, which typically incorporates random negative sampling to address implicit feedback scenarios, fails to fully address **the unique challenges inherent in implicit feedback**. In our view, this area presents significant opportunities for advancement.
>
> In contrast, our work specifically addresses Cross-Domain Recommendation from Implicit Feedback (CDRIF). We introduce an innovative framework tailored to this particular issue. This framework, designed to be compatible with existing techniques, has demonstrated its effectiveness through rigorous experiments. Its model-agnostic and plug-and-play nature facilitates easy implementation and integration into current systems, a significant advantage over existing methods.
>
> Addressing the critique of our technical contribution, it is important to note the distinctiveness of our framework. Developing a versatile framework compatible with a range of existing techniques, especially for CDRIF, is a non-trivial task. Our work pioneers in this area, marking a significant technical advancement. **Therefore, dismissing our technical contribution in designing this innovative framework from scratch would be unjust**.
>
> Our framework's uniqueness is further highlighted by its ability to address two critical challenges in recommendation systems:
>
> 1. **Cold-Start Problem in Implicit-Feedback Systems**: Traditional implicit-feedback systems struggle with this issue. Our framework, as evidenced in our experiments, effectively leverages knowledge transfer from a source domain to mitigate the cold-start problem in the target domain.
>
> 2. **Handling Implicit Feedback in CDR Methods**: Current CDR methods inadequately address implicit feedback, often resorting to a combination of traditional CDR methods and negative sampling. Our framework, however, goes beyond this by systematically investigating cross-domain recommendations under implicit feedback scenarios and proposing a comprehensive solution rather than just methodological adjustments.
>
> In summary, our paper is the **first** to thoroughly explore CDR under implicit feedback, proposing a robust and adaptable framework for CDRIF. This framework not only outperforms existing baselines with **a large margin** but also offers ease of implementation and deployment in practical settings. **Therefore, again, dismissing our contribution to designing a novel, general framework from scratch would overlook the substantial advancements our work presents in this field**.
>
> > Baseline methods are not well selected.
>
> **Response 2:** We understand your concern regarding the baseline methods selection. In our study, we focused on methods most relevant to our specific research question. However, we acknowledge the omission of certain important works in implicit feedback recommendation systems. We agree that including these methods could provide a more comprehensive understanding. To address this, we are actively expanding our experimental scope to include these important methods. We are committed to ensuring a thorough and robust evaluation of our framework.
>
> **We will promptly report the updated results once these additional experiments are completed**.

---

> > ### Author Response · Authors · 2023-11-20
> > **Reply - Part II**
> >
> > > The presentation of this work needs improvement.
> >
> > **Response 3:** We appreciate your feedback on the clarity of our presentation. We plan to revise the paper to simplify notations and provide a clearer explanation of our framework.
> >
> > > Why the learnable calibration parameter can help denoising? What's the difference between this idea and the active learning idea?
> >
> > **Response 4:** **It seems there may be some confusion regarding our calibration parameter.** I would like to clarify that this parameter in our model is not learnable through training. Instead, it is a predefined factor in the NARF model, which has demonstrated its effectiveness in our experiments, particularly in enhancing recommendation accuracy. The primary purpose is to calibrate the training data to better reflect users’ actual preferences. Specifically, it involves determining the calibration factor for each user-item pair, with a higher value indicating closer alignment with the user’s genuine preferences. Based on these factors, we can prioritize training the recommender system with data that carry higher factors.
> >
> > Our approach differs from traditional active learning, particularly in how we handle outliers and noise. Traditional active learning is sensitive to outliers and noise [r1]. To mitigate this vulnerability, our framework incorporates adaptive denoising strategies. These strategies are designed to effectively filter out noise and reduce the influence of outliers. By integrating these denoising approaches, we enhance the robustness of our recommendation system, leading to improved performance and more reliable recommendations. This integration of adaptive denoising represents a key advancement over traditional active learning methods, addressing a critical challenge and boosting the overall efficacy of our recommendation framework.
> >
> > > What's the performance comparison with the state-of-the-art negative sampling method in learning to rank?
> >
> > **Response 5:** We appreciate the emphasis on comparing our work with advanced negative sampling methods. While our current study did not encompass this specific analysis, we recognize this as a limitation and will consider incorporating it.
> >
> > However, it is important to note that our primary focus is not on negative sampling methods. The prevalent approaches in CDRIF predominantly rely on random negative sampling. Our framework builds upon this foundation. We have found that even when employing the most basic sampling method, such as random negative sampling, CDR algorithms that utilize our framework still manage to achieve commendable results.
> >
> > This observation underscores the inherent strength and versatility of our framework, suggesting that its effectiveness is not solely dependent on advanced negative sampling techniques. Nevertheless, a comparative analysis with negative sampling methods could provide additional insights, and we plan to explore this in our subsequent work.
> >
> > **In conclusion**, we thank you for your constructive comments and will strive to improve our work accordingly and finally update a new version of our paper here (we are running some experiments now and will try our best to update our paper at the end of the author-reviewer discussion). We do hope to get your new responses for our updated paper, and **your re-evaluation is very important for us (for this time or future).**
> >
> > [r1] Han, B., Yao, Q., Yu, X., Niu, G., Xu, M., Hu, W., ... & Sugiyama, M. (2018). Co-teaching: Robust training of deep neural networks with extremely noisy labels. Advances in neural information processing systems, 31.

---

> > > ### Author Response · Authors · 2023-11-22
> > >
> > > Dear Reviewer LrYi,
> > >
> > > I hope this message finds you well. I am writing to provide an update on the additional experiments we have conducted in response to the feedback. As suggested, we have included comparative results with two additional baselines: DCDCSR (Zhu, Feng, et al. A deep framework for cross-domain and cross-system recommendations. In IJCAI 2018.) and LTR (Chen, Jin, et al. Learning Recommenders for Implicit Feedback with Importance Resampling. Proceedings of the ACM Web Conference 2022.). As LTR is not the algorithm for CDR, we incorporate it with DCDCSR as DCDCSR-LTR.
> > >
> > > The results of these experiments are as follows:
> > >
> > > | **Metric** | **DCDCSR** | **DCDCSR-LTR** | **Ours** |
> > > |:----------:|:----------:|:---------:|:----------:|
> > > | **R@50**   | 0.0058     | 0.0575    | **0.1268** |
> > > | **R@100**  | 0.0098     | 0.0721    | **0.1767** |
> > > | **N@50**   | 0.0042     | 0.0512    | **0.1066** |
> > > | **N@100**  | 0.0053     | 0.0589    | **0.1275** |
> > >
> > > These results demonstrate the significant improvement our method offers over the aforementioned baselines. We believe this additional evidence further strengthens our method. We hope this addresses any concerns and provides a clearer understanding of our work's contribution. In this regard, it would be greatly appreciated if you could take a moment to provide any further feedback or comments on our rebuttal response. We understand the time and effort involved in this process and are grateful for your continued guidance and expertise.
> > >
> > > Thank you once again for your time and consideration. We look forward to any further thoughts you might have.

---

> > ### Comment · Reviewer_LrYi · 2023-11-22
> >
> > Thanks for the response.
> >
> > The reviewer agrees that this framework is compatible with existing techniques, however, this still cannot address the concerns on technical novelty. All the proposed components have been deeply investigated in recommendation with implicit feedback. This work can only be viewed as a specific use case in cross-domain recommendation.
> >
> > "Cold-Start Problem in Implicit-Feedback Systems: Traditional implicit-feedback systems struggle with this issue. Our framework, as evidenced in our experiments, effectively leverages knowledge transfer from a source domain to mitigate the cold-start problem in the target domain."
> >
> > A: This is not convincing. Many work has been proposed to solve cold start problem in recommendation with implicit feedback. The reviewer suggests the authors to re-do the literature survey.
> >
> > "Handling Implicit Feedback in CDR Methods: Current CDR methods inadequately address implicit feedback, often resorting to a combination of traditional CDR methods and negative sampling. Our framework, however, goes beyond this by systematically investigating cross-domain recommendations under implicit feedback scenarios and proposing a comprehensive solution rather than just methodological adjustments."
> >
> > A: From above words, the reviewer still cannot see what's the specific contribution and particular adaptation proposed in this framework. There are several existing works about CDR using implicit data, and the components from this framework are not novel as well.

---

### Meta-Review · Area_Chair_2xPK · 2023-12-06

**Metareview:**

(a) Summarize the scientific claims and findings of the paper based on your own reading and characterizations from the reviewers.
- Cross-domain recommender systems from implicit feedback in both source and target domains is challenging is largely unexplored
- The authors propose a three-step framework for sampling, calibrating, and denoising implicit data.

(b) What are the strengths of the paper?
- The paper works on an interesting and valuable problem
- The proposed framework is sensible and the empirical study shows its potential

(c) What are the weaknesses of the paper? What might be missing in the submission?
- The current manuscript could improve its positioning with respect to current literature (and notably discuss and compare to other recent relevant works). Currently, some of its claims with respect to prior work and novelty appear unjustified.
- The proposed method, in each of its three steps, reuses well-known ideas from the literature. Its value would be more clearly demonstrated by a thorough empirical validation (e.g., with a full set of current baselines).

**Justification For Why Not Higher Score:**

The original reviews discussed several significant limitations of the current version of this work. The reviewers' assessments were relatively well-supported, and the inter-reviewer agreement was high.

The authors did provide a response that brought some clarifications. The reviewers acknowledged the authors' reply (both through public and private comments), but their initial assessment remained. Overall, demonstrating empirically the usefulness of the proposed approach might greatly improve this work.

**Justification For Why Not Lower Score:**

N/A

---

### Decision · Program_Chairs · 2024-01-16

Reject